# Deciphering the chemical language of inbred and wild mouse conspecific scents

Maximilian Nagel[1‡], Marco Niestroj[1], Rohini Bansal[2], David Fleck[1], Angelika Lampert[3,4], Romana Stopkova[5], Pavel Stopka[5]*, Yoram Ben-Shaul[2]*, Marc Spehr[1,4]*§

[1]Department of Chemosensation, Institute for Biology II, RWTH Aachen University, Aachen, Germany; [2]Department of Medical Neurobiology, Institute for Medical Research Israel Canada, Faculty of Medicine, The Hebrew University of Jerusalem, Jerusalem, Israel; [3]Institute of Neurophysiology, Uniklinik RWTH Aachen University, Aachen, Germany; [4]Research Training Group 2416 MultiSenses – MultiScales, RWTH Aachen University, Aachen, Germany; [5]BIOCEV group, Department of Zoology, Faculty of Science, Charles University, Prague, Czech Republic

*For correspondence:
pavel.stopka@natur.cuni.cz (PS);
yoramb@ekmd.huji.ac.il (YBS);
m.spehr@sensorik.rwth-aachen.de (MS)

Present address: ‡Sensory Cells and Circuits Section, National Center for Complementary and Integrative Health, Bethesda, United States

§Lead contact

Competing interest: The authors declare that no competing interests exist.

**Abstract** In most mammals, conspecific chemosensory communication relies on semiochemical release within complex bodily secretions and subsequent stimulus detection by the vomeronasal organ (VNO). Urine, a rich source of ethologically relevant chemosignals, conveys detailed information about sex, social hierarchy, health, and reproductive state, which becomes accessible to a conspecific via vomeronasal sampling. So far, however, numerous aspects of social chemosignaling along the vomeronasal pathway remain unclear. Moreover, since virtually all research on vomeronasal physiology is based on secretions derived from inbred laboratory mice, it remains uncertain whether such stimuli provide a true representation of potentially more relevant cues found in the wild. Here, we combine a robust low-noise VNO activity assay with comparative molecular profiling of sex- and strain-specific mouse urine samples from two inbred laboratory strains as well as from wild mice. With comprehensive molecular portraits of these secretions, VNO activity analysis now enables us to (i) assess whether and, if so, how much sex/strain-selective 'raw' chemical information in urine is accessible via vomeronasal sampling; (ii) identify which chemicals exhibit sufficient discriminatory power to signal an animal's sex, strain, or both; (iii) determine the extent to which wild mouse secretions are unique; and (iv) analyze whether vomeronasal response profiles differ between strains. We report both sex- and, in particular, strain-selective VNO representations of chemical information. Within the urinary 'secretome', both volatile compounds and proteins exhibit sufficient discriminative power to provide sex- and strain-specific molecular fingerprints. While total protein amount is substantially enriched in male urine, females secrete a larger variety at overall comparatively low concentrations. Surprisingly, the molecular spectrum of wild mouse urine does not dramatically exceed that of inbred strains. Finally, vomeronasal response profiles differ between C57BL/6 and BALB/c animals, with particularly disparate representations of female semiochemicals.

## eLife assessment

This carefully executed study provides a comparison of the chemical composition of mouse urine across strain and sex with the responses of vomeronasal sensory neurons, which are responsible for detecting chemical social cues. While the authors did not examine all molecular classes found in mouse urine or directly test whether the urinary volatile chemicals that vary with sex and strain are effective vomeronasal neuron ligands, **solid** data are provided that will be of significant interest to those studying chemical communication in rodents. This work should provide a **valuable** foundation

for future research that will determine which molecules drive sex- and strain-specific vomeronasal responses.

## Introduction

In rodents and most other mammals, the accessory olfactory system controls conspecific chemical communication during social interactions (*Brennan and Zufall, 2006*; *Dulac and Torello, 2003*; *Mohrhardt et al., 2018*; *Tirindelli et al., 2009*). Ethologically meaningful chemostimuli that trigger stereotypic social and sexual behaviors are predominantly detected by the vomeronasal organ (VNO), a tubular sensory structure at the anterior base of the nasal septum. In a crescent-shaped medial neuroepithelium, the VNO harbors approximately 100,000–200,000 vomeronasal sensory neurons (VSNs; *Wilson and Raisman, 1980*), each extending a single unbranched apical dendrite that terminates in a paddle-shaped swelling (*Mohrhardt et al., 2018*). At the dendritic tips, microvilli are immersed in mucus that fills a central luminal canal, which extends via the vomeronasal duct into the nasal cavity. During social investigatory behavior, which in mice primarily involves periods of intense licking and sniffing of both facial and anogenital regions (*Luo et al., 2003*), semiochemicals are sucked into the VNO lumen. Upon binding to cognate vomeronasal receptors (VRs), the chemical signal is transduced into electrical VSN activity and, ultimately, neuronal discharge.

Behaviorally relevant natural chemostimuli are, typically, complex blends of compounds (*Wyatt, 2017*) in various bodily secretions (*Albone, 1984*). By far, the most widely studied secretion in animal chemosensory research is urine (see *Mohrhardt et al., 2018*, and references therein), which is a rich source of semiochemicals that serves a well-established function in social communication. While we still lack a comprehensive molecular description of this broadband vomeronasal stimulus, previous work has identified several putative semiochemicals in mouse urine and other bodily secretions, which activate VSNs and cover many structural groups and feature dimensions (*Chamero et al., 2007*; *Doyle et al., 2016*; *Fu et al., 2015*; *Hurst et al., 2001*; *Kimoto et al., 2005*; *Leinders-Zufall et al., 2004*; *Leinders-Zufall et al., 2000*; *Nodari et al., 2008*; *Novotny, 2003*; *Overath et al., 2014*; *Rivière et al., 2009*; *Röck et al., 2006*; *Sturm et al., 2013*; *Wyatt, 2017*). Prominent molecularly identified VSN stimuli include various sulfated steroids (*Celsi et al., 2012*; *Fu et al., 2015*; *Haga-Yamanaka et al., 2015*; *Haga-Yamanaka et al., 2014*; *Isogai et al., 2011*; *Nodari et al., 2008*; *Turaga and Holy, 2012*), which could reflect the dynamic endocrine state of an individual.

So far, virtually all research on vomeronasal physiology is based on urinary stimuli derived from inbred laboratory mice (but see *Bansal et al., 2021*). This facilitates standardization across studies but, while VSN sex and strain selectivity has previously been explored (*Fu et al., 2015*; *He et al., 2008*), it remains unclear whether secretions collected from inbred mice accurately represent the potentially more ethologically relevant stimuli found in the wild. In fact, while wild *Mus musculus* populations exhibit several-fold higher levels of genetic variation than those in human populations, commonly studied inbred strains of laboratory mice derive from a limited set of founders. Therefore, such strains contain only a small subset of the genetic variation that is present in nature (*Phifer-Rixey and Nachman, 2015*). Classical inbred strains of house mice are genetic mosaics of the three main wild subspecies, *M. m. domesticus*, *musculus*, and *castaneus*, which started to diverge ~350,000–500,000 years ago. However, the genomes of prominent inbred strains, such as C57BL/6 or BALB/c, are predominantly derived from *M. m. domesticus* (*Wade and Daly, 2005*). The mitochondrial genomes of both laboratory strains are identical, implying a common descent along the maternal line. Given their overlapping geographical distribution, wild house mice subspecies have also undergone secondary contact and hybridization (*Duvaux et al., 2011*), which has diversified their genetic landscape.

Laboratory inbreeding, which has been ongoing for more than 100 years for C57BL/6 and BALB/c strains (*Phifer-Rixey and Nachman, 2015*), may have affected the chemical composition of bodily secretions and, consequently, their information content. Indeed, both within and between strains, laboratory mice lack variation in major urinary protein (MUP) patterns typically found among wild mice (*Cheetham et al., 2009*). This inbred homogeneity could have important implications for research investigating, for example, social recognition or mate choice: if there are marked qualitative and/or quantitative differences in chemical composition of secretions and, accordingly, in the neuronal representations of wild as compared to inbred derived stimuli, one might question the conclusions based on a large body of work using inbred secretions (*Bansal et al., 2021*).

On the recipients' end, the level of chemosensory information available from con/heterospecific secretions is determined by individual VR expression profiles. Across strains, VR repertoires vary as a function of both genetics and experience (*Duyck et al., 2017*; *Ibarra-Soria et al., 2017*; *Wynn et al., 2012*; *Xu et al., 2016*). Therefore, it is unclear whether receptor arrays in laboratory strains, despite hundreds of generations of inbreeding and domestication ('microevolution') in a laboratory environment, have retained selectivity for more ethologically relevant wild-derived stimuli. Moreover, it remains uncertain whether VR tuning profiles enable VSNs to capture key ethological features from molecular concentration differences between sex- and strain-specific secretions (*He et al., 2010*; *Leinders-Zufall et al., 2000*). Together, four key biological questions arise from these lines of reasoning: (i) Do VSN response profiles reflect the global molecular content of urine (i.e. are these neurons sensitive to many/most compounds) or, by contrast, is the VNO a highly selective molecular detector (responding to just a few select molecules)? (ii) Are VSN response profiles strain-specific? (iii) Which semiochemicals provide information about sex and/or strain? (iv) Is there something unique about wild mouse secretions and/or the VSN response profiles they trigger?

Here, to address these unresolved issues, we combine a robust VSN activity assay with comparative molecular profiling of sex- and strain-specific mouse urine from two inbred laboratory strains and wild mice. Our study provides molecular portraits of these secretions. We report that large fractions of generic urine compounds are shared among both male and female mice of all genetic backgrounds. We further show that the urinary 'secretome' in wild mice does not differ dramatically from that found in laboratory strains. Surprisingly, while male urine contains much higher protein concentrations, females secrete a larger variety of proteins (including MUPs). For proteins common to all strains, concentrations are relatively low in C57BL/6, moderate in BALB/c, and high in wild animals. However, despite this concentration bias, there is no overrepresentation of wild-selective proteins. Notably, both the volatile organic compound (VOC) and protein profile of urine, each provides sufficient information to decode a given sex/strain combination (with protein content exhibiting stronger discriminative power). Moreover, we identify a rich lipocalin repertoire in urine, which alone could allow chemosensory discrimination of sex/strain combinations.

A key strength of this study is the use of the exact same stimuli as previously employed to investigate sensory representations in the accessory olfactory bulb (AOB) (*Bansal et al., 2021*), the first central processing stage along the accessory olfactory pathway. Our previous work demonstrated that AOB representations of ethologically relevant urine stimuli are similar for male mice from two different inbred strains (C57BL/6 and BALB/c), despite potential differences in VR repertoires. In addition, we found that wild mouse stimuli elicit responses that, although not identical, are nevertheless qualitatively similar to those from commonly used inbred strains (*Bansal et al., 2021*). VSN activity analysis now enables us to ask whether these features also manifest in the VNO and, thus, to assess (i) whether the information inherent in a sex- and strain-specific urinary secretome is accessible via vomeronasal sampling; and (ii) if, on the population level, VSN sensory representations differ between strains. Comparing responses from male C57BL/6 and BALB/c mice (as done previously on AOB level; *Bansal et al., 2021*), our data demonstrate that recipient strain identity is reflected by their VSN activity patterns. Moreover, when exposed to stimuli from different strains, large VSN fractions (often >50%) respond to only one stimulus, suggesting a substantial degree of selective sampling. Together, our study reveals selective and strain-dependent representations of urine chemical content, a surprising scarceness of selective responses to wild stimuli, as well as remarkably rich and sex/strain-specific molecular profiles that likely preserve most of the biologically relevant information.

## Results

In mice, the VNO is the predominant sensory structure mediating conspecific chemical communication. In this study, by comparing samples derived from two inbred strains (BALB/c and C57BL/6) as well as from wild mice, we pursue four main questions: (i) Which chemical components in mouse urine – a major source of semiochemicals – distinguish sex and/or strain? (ii) To what degree has inbreeding affected the chemical composition of urine (i.e. how unique are wild mouse secretions)? (iii) How much of the sex- and strain-specific chemical information in urine is accessible to a conspecific via vomeronasal sampling (i.e. how selective are VSN response profiles)? (iv) Upon stimulation with the same stimuli, do VSN response patterns differ between strains?

## A 'low noise' assay to capture VSN population activity

To record VSN signal fingerprints in response to naturalistic stimuli, it is essential to establish a robust population activity assay that reliably captures the raw information content inherent in bodily secretions. To this end, we analyzed single-cell $Ca^{2+}$ transients among large VSN populations in acute coronal VNO sections (*Figure 1a and b*). Our experimental design followed a standard pairwise comparison paradigm (*Figure 1c*). Pairs of pooled stimuli differed either in the donors' sex (male *versus* female) or in their genetic background (BALB/c, C57BL/6, or wild). We repeated brief (10 s) alternating stimulus presentations twice at 180 s inter-stimulus intervals that ensured recovery from VSN adaptation (*Wong et al., 2018*). Experiments concluded with a brief exposure to elevated extracellular $K^+$ ($S_3$) to depolarize neurons and test for integrity of each neuron's spike generation machinery. Dependent on their individual urine response profiles, VSNs were categorized as either specialists (selective response to one stimulus) or generalists (responsive to both stimuli). Overall, we recorded >43,000 $K^+$-sensitive neurons, of which a total of 16,715 VSNs (38.4%) responded to urine stimulation. Of these urine-sensitive neurons, 61.4% displayed generalist profiles, whereas 38.6% were categorized as specialists (*Figure 1c and d*). As a measure of intrinsic signal variability, we calculated reliability indices that quantify the similarity (or lack thereof) of two successive responses to the same stimulus (with small values reflecting high reliability; see Materials and methods). We analyzed both $Ca^{2+}$ signal amplitudes (*Figure 1d*) and integrals (*Figure 1—figure supplement 1a*) as measures of response magnitude. For both generalist and specialist VSNs, response reliability indices are normally distributed around zero. While the distribution of reliability indices derived from response integral measurements is somewhat broad (*Figure 1—figure supplement 1a*), indices based on signal amplitudes proved relatively homogeneous (*Figure 1d*). Accordingly, we use average response amplitudes as indicators of signal strength throughout this study.

In a first set of control experiments, we asked how VSNs from male C57BL/6 mice respond when challenged with urine samples from two groups of animals of the same sex and inbred strain (*Figure 1—figure supplement 1b and c*). Since the chemical composition of both stimuli should be similar, we expected that the vast majority of urine-sensitive VSNs will display generalist response profiles. That was indeed the case. Less than 2% of VSNs showed stimulus selectivity (*Figure 1—figure supplement 1b and c*). Moreover, response indices (reflecting a bias toward either of the paired stimuli) are normally (and narrowly) distributed around zero (*Figure 1—figure supplement 1b and c*). Both observations confirm a low level of biological and experimental noise in this setting. Thus, our assay is well suited to detect and compare VSN sensory responses upon pairwise stimulation.

## Sex-specific stimuli elicit distinct VSN sensory representations

We next investigated how sex differences are reflected in VSN response profiles. When challenged with male *versus* female urine from C57BL/6 animals, the fraction of specialist neurons more than doubled to 5% of all $K^+$-sensitive neurons. Notably, female urine recruited more specialist neurons than male urine (*Figure 1e*). We then asked whether this pattern is correlated with the chemical compositions of male and female urine. In-depth molecular analysis of urine content via two-dimensional gas chromatography–mass spectrometry (GCxGC-MS) as well as nanoliquid chromatography-tandem mass spectrometry (nLC-MS/MS) identified a total of 1006 molecules (detected in ≥3 out of 10 male or female samples, respectively), of which approximately 40% are low molecular weight VOCs, while 60% are proteins. Roughly half of the molecules in either group are found in both male and female urine (*Figure 1e*). Unexpectedly, while we hardly identify any male-specific proteins in C57BL/6 urine, we find a large fraction of female-specific proteins. We asked whether this phenomenon is (i) a distinct feature of C57BL/6 mice, (ii) common among inbred laboratory strains, or (iii) also observed in wild animals. Therefore, we included urine samples from both BALB/c and wild mice in extended molecular profiling. In both groups, we observed similarly increased levels of female-specific proteins (*Figure 1f*). The total amount of protein, however, is substantially enriched in male urine (*Figure 1g*). Our data thus suggest that, while females secrete a larger variety of proteins, overall concentrations are comparatively low.

Finally, we asked whether common compounds (i.e. molecules identified in both male and female urine) show concentration disparities between sexes and, if so, how such differences are reflected in VSN response profiles. When VSNs were challenged with male *versus* female C57BL/6 urine, 19% of all $K^+$-sensitive neurons responded to both stimuli (*Figure 1e*). Here, again, response preference indices

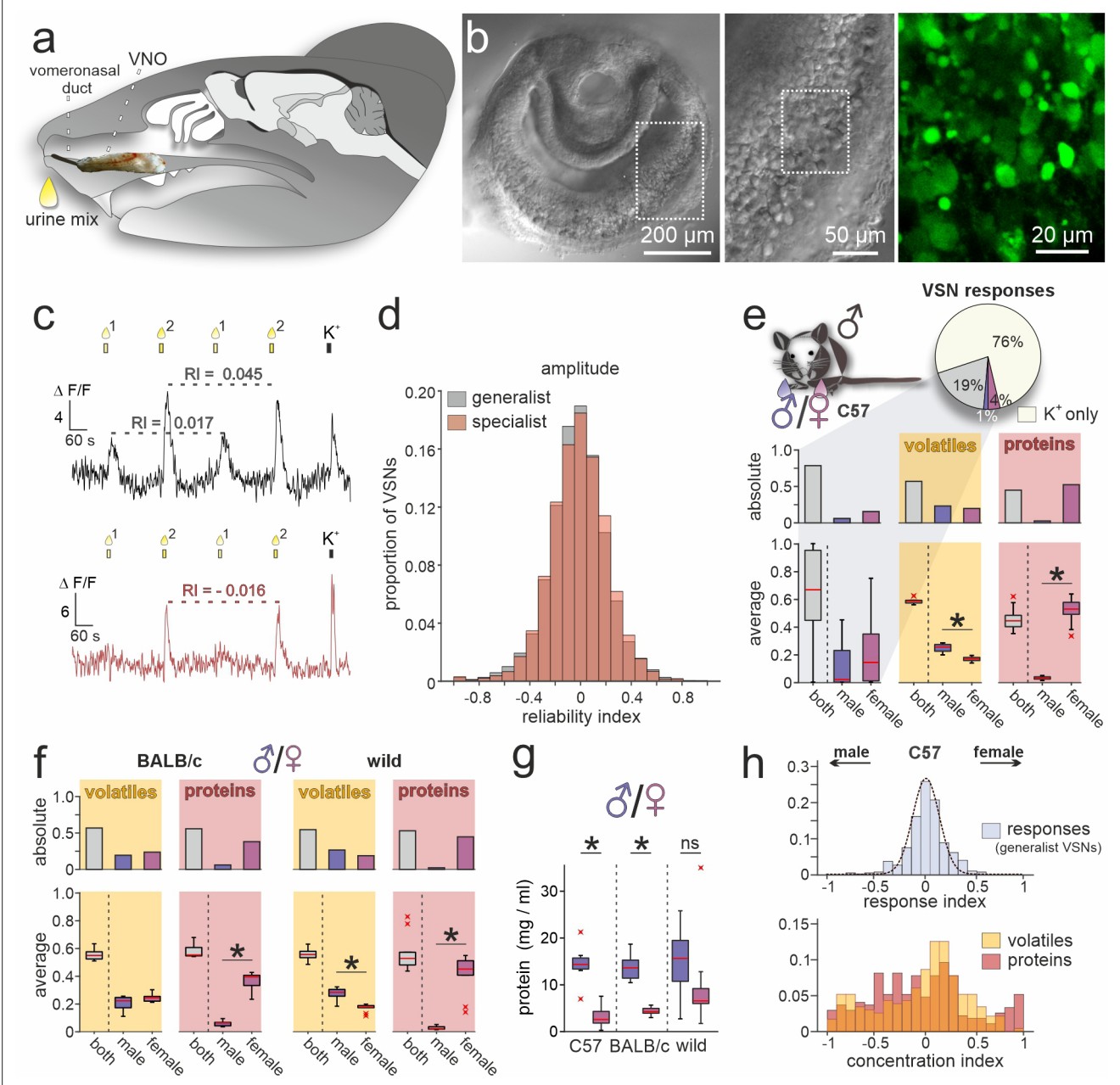

**Figure 1.** A population activity assay captures sex-dependent stimuli representations. (**a**) Anatomical location of the rodent vomeronasal organ (VNO). Schematic depicting a sagittal section through a mouse head with an overlay of a VNO image. The vomeronasal duct that opens into the anterior nasal cavity is also highlighted. (**b**) Overview (left) and zoomed-in (middle) differential interference contrast micrograph showing an acute coronal VNO section from an adult C57BL/6 mouse. Confocal fluorescence image (right; dashed rectangle in middle) depicting a depolarization (elevated $K^+$) dependent cytosolic $Ca^{2+}$ increase in vomeronasal sensory neuron (VSN) somata after bulk loading with Cal-520 AM. (**c**) Original traces showing changes in cytosolic $Ca^{2+}$ concentration over time in two representative VSN somata. VNO slices were briefly challenged with two mixtures of diluted mouse urine (1:100; 10 s; top yellow bars/droplets). Repeated stimulation in alternating sequence at 180 s inter-stimulus intervals (***Wong et al., 2018***) was followed by membrane depolarization upon exposure to elevated extracellular $K^+$ (50 mM; 10 s). According to individual response type, VSNs were categorized as 'generalists' (top) or 'specialists' (bottom). Signal amplitudes in response to the same stimulus allowed calculation of a reliability index (RI) as a measure of signal robustness. (**d**) Amplitude reliability index histograms of all generalist VSNs (gray; n=10,258) and all specialist VSNs (red; n=6457) recorded in this study. Note that for both response types, indices are normally distributed with a narrow central peak around zero. (**e**) Quantification of results obtained from recordings in VSNs from male C57BL/6 mice challenged with male *versus* female C57BL/6 urine stimuli. Pie chart (top) illustrates the proportions of generalist (19%) and specialist neurons (1% and 4%, respectively) among all $K^+$-sensitive VSNs (n=1999). Bar graph (middle, left) breaks down the summed total of urine-sensitive neurons by categories and compares their distribution with the proportions of volatile organic compounds (VOCs) (yellow background; n=405) and proteins (red background; n=601) found either in both male and female urine (gray bars) or exclusively in

*Figure 1 continued on next page*

*Figure 1 continued*

samples from one sex (male, dark blue; female, purple). Box-and-whisker plots (bottom, left) illustrating generalists-to-specialists ratios over individual experiments (n=19). Boxes represent the first-to-third quartiles. Whiskers represent the 10th and 90th percentiles, respectively. Outliers (1.5 IQR; red x) are plotted individually. The central red band represents the population median ($P_{0.5}$). Results are shown in relation to box-and-whisker plots that outline chemical content data obtained from paired comparisons (n=10 individuals per group). (**f**) Molecular composition of male *versus* female urine from BALB/c and wild mice. Bar graphs (top) display proportions of VOCs (yellow background; BALB/c, n=514; wild, n=462) and proteins (red background; BALB/c, n=407; wild, n=526) found either in both male and female urine (gray bars) or exclusively in samples from one sex (male, dark blue; female, purple). Box-and-whisker plots (bottom) quantify category ratios for individual paired experiments (n=10 individuals per group). (**g**) Quantification (Bradford assay) of protein/peptide content in urine samples from C57BL/6, BALB/c, and wild mice, respectively (n=10 each). Note the substantially increased protein content in male samples. (**h**) Response index histogram (top) obtained from generalist C57BL/6 VSNs that responded to both male and female same-strain urine (data corresponding to (**e**)). Fitted Gaussian curve (dashed line) centers close to zero (peak = –0.03) and shows a relatively narrow width (σ=0.13). By contrast, concentration index histograms (bottom), calculated for VOCs (yellow) and proteins (red) found in both male and female urine samples, are heterogeneous and not normally distributed. Asterisks (*) indicate statistical significance, p<0.05; Wilcoxon signed-rank test (for VSN functional data), Mann-Whitney U test (for molecular profiling), and unpaired t-test (for protein content comparison shown in (**g**)).

The online version of this article includes the following figure supplement(s) for figure 1:

**Figure supplement 1.** A VSN population activity assay with a low level of biological and experimental noise.

are narrowly distributed and center around zero (*Figure 1h*). By contrast, analogously calculated concentration indices (see Materials and methods) that can reflect potential disparities are distributed more broadly and non-normally (*Figure 1h*). This apparent incongruence between nonpreferential generalist sensitivity and relatively large chemical concentration differences indicates that VSN responses (at least in C57BL/6 males) do not simply reflect the information theoretically available via concentration differences between male and female same-strain stimuli. One explanation for this is that common ligands are not well represented by the observed general concentration disparities. Alternatively, it may be that even low concentrations fully activate a given VSN, and thus concentration differences are not reflected by response strengths. Notably, a broad and non-normal distribution of concentration indices among molecular components of male *versus* female urine is also found in samples from both BALB/c and wild mice (*Figure 1—figure supplement 1d*).

## Chemical characterization of urine content identifies distinctive molecular signatures of sex and strain

Which differences in chemical composition (i.e. which molecules) characterize sex/strain-specific secretions and may thus provide information about sex and strain to a recipient? Molecular profiling of urine content provides a unique opportunity to address this question and identify enriched compounds (i.e. present in ≥6 of 10 individual samples) with the potential capacity to signal an animal's sex, strain, or both. Adopting this more conservative criterion, we detected 208 enriched VOCs and 264 enriched proteins. Almost half of all VOCs (47.6%) and about one-third of all proteins (31.4%) are found in all six sex/strain combinations tested and are, thus, considered generic mouse urine components (*Figure 2a and b*). Moreover, a total of 33 compounds (10 VOCs, 23 proteins) were identified across all strains, but in a sex-specific fashion. Similarly, 22 molecules (10 VOCs, 12 proteins) revealed strain selectivity, independent of sex. Notably, a large fraction of compounds, i.e., 34.1% of all VOCs (71/208) and 22.3% of proteins (59/264), were detected exclusively in one of the six sex/strain combinations. Only 18 of the 208 VOCs (8.7%) could not be categorized as either generic or specific (for either sex or strain, or a unique sex/strain combination). Among the 82 proteins that showed sex specificity, either across strains (23 proteins) or as part of a unique sex/strain combination (59 proteins), the vast majority (89.0%) was found in female samples. For VOCs, however, the opposite picture emerged. Here, 52 of 81 sex-specific compounds (64.2%) were selectively detected in male samples. Together, chemical profiling revealed (i) that large fractions of urine content are shared among both laboratory and wild mice; (ii) that roughly one-third of urinary VOCs and one-fourth of proteins are exclusively found in a given sex/strain combination; and (iii) that male and female mice might have adopted different chemical secretion strategies to signal their sex.

If information coding along the accessory olfactory pathway would strictly follow a 'labeled-line' logic (*Ishii et al., 2017*), absence or presence of a given molecule could be adequate to signal sex or strain. Several recent lines of evidence, however, suggest a combinatorial coding strategy that also involves some level of circuit plasticity (*Kaur et al., 2014*; *Mohrhardt et al., 2018*; *Xu et al., 2016*).

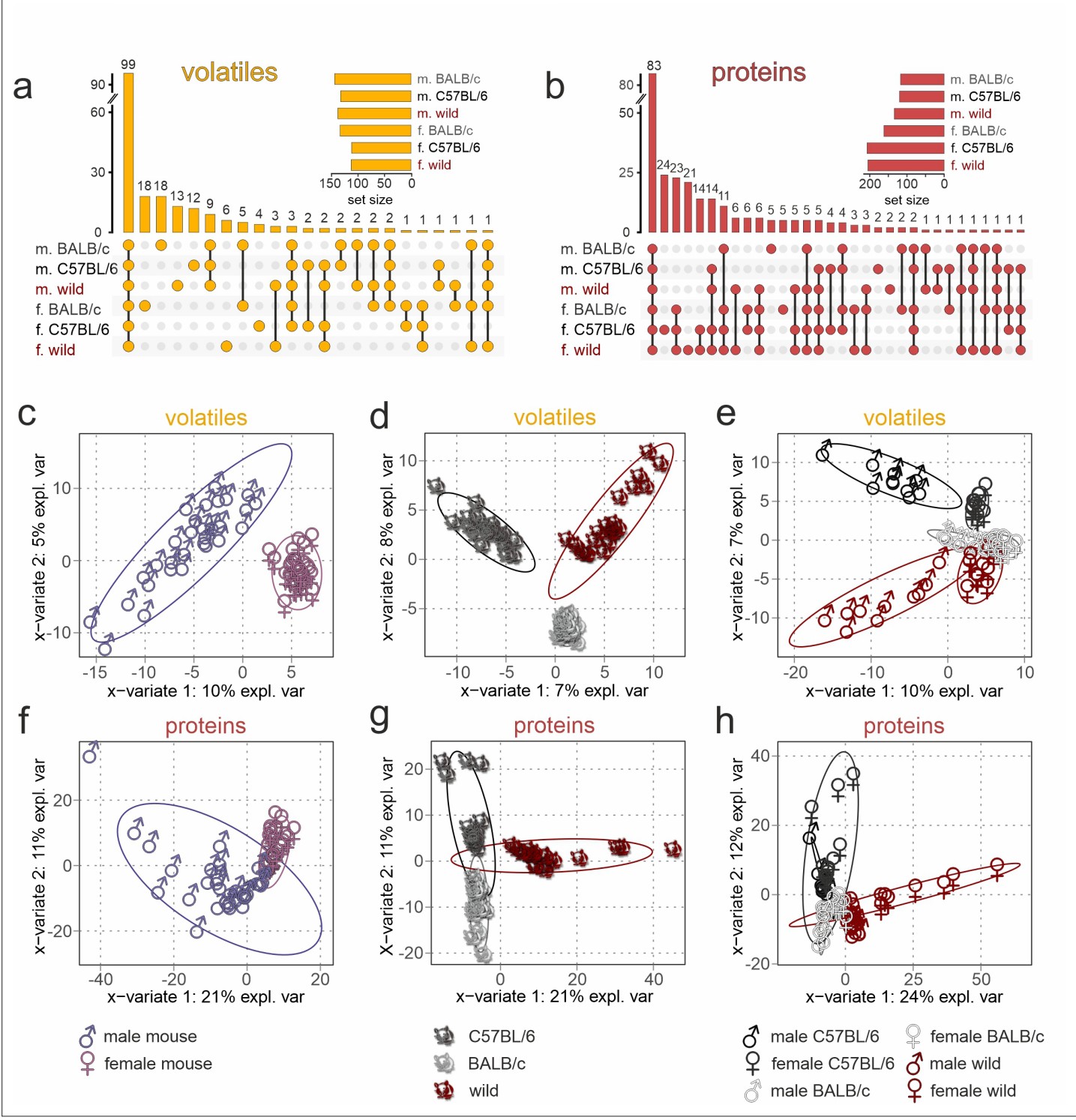

**Figure 2.** Chemical profiling of urine content identifies unique sex- and strain-specific molecular fingerprints. (**a, b**) Matrix layout for all intersections of volatile organic compounds (VOCs) (**a**) and proteins (**b**) among the six sex/strain combinations, sorted by size. Colored circles in the matrix indicate combinations that are part of the intersection. Bars above the matrix columns represent the number of compounds in each intersection. Empty intersections have been removed to save space. Horizontal bar charts (bottom, left) depict the number of VOCs (**a**) and proteins (**b**) detected in each urine set. Proteomics data are available via ProteomeXchange with identifier PXD042324. (**c–h**) Sparse partial least-squares discriminant analysis (sPLS-DA) score plots depicting the first two sPLS-DA components, which explain 7–10% (1st component) and 5–8% (2nd component) of VOC data variance (**c–e**) as well as 21–24% (1st component) and 11–12% (2nd component) of protein data variance (**f–h**), respectively. Ellipses represent 95% confidence

*Figure 2 continued on next page*

*Figure 2 continued*

intervals. Plots demonstrate sample clustering according to the urine donors' sex (**c, f**), genetic background (**d, g**), or sex/strain combination (**e, h**). Each data point represents a sample from an individual animal (n=60; 10 samples per sex/strain combination), with sample type colored according to symbol legend (bottom).

The online version of this article includes the following figure supplement(s) for figure 2:

**Figure supplement 1.** The most informative VOCs and proteins that discriminate sex and strain, respectively.

**Figure supplement 2.** Protein and VOC content in mouse urine across strains.

**Figure supplement 3.** Hierarchical clustering of mouse urine lipocalin content reveals an unexpectedly diverse repertoire of 27 lipocalins.

Therefore, we next asked if the total VOC or protein content of a given urine sample preserves sufficient predictive/discriminative information to classify samples according to sex, strain, or a specific sex/strain combination. We used sparse partial least-squares discriminant analysis (sPLS-DA) (*Lê Cao et al., 2011*), a chemometrics machine learning technique, to reduce data dimensionality and optimize sample separation (*Gromski et al., 2015*). When plotted on two-dimensional coordinates that represent the most discriminative variables (*Figure 2c–h*), both VOCs (*Table 1*) and proteins provide sufficient information to cluster stimuli according to sex (*Figure 2c and f*), strain (*Figure 2d and g*), or a combination of both variables (*Figure 2e and h*). We then calculated variable importance in projection scores as measures of a particular variable's informative power, which is correlated to the variance explained by the model (*Gromski et al., 2015*; *Lê Cao et al., 2011*). Generally, we find that protein content exhibits stronger discriminative power (21–24% and 11–12% of explained variance) than VOC content (7–10% and 5–8% of explained variance).

We next aimed to identify the most relevant variables (i.e. molecules) for sex or strain classification. Training Random Forest classifiers (*Breiman, 2001*), we obtained feature significance scores, the 'Gini importance' (*Breiman, 2001*), that provide relative relevance rankings of the individual variables. Here, we list the 20 most informative VOCs (*Table 2*) and proteins that discriminate sex and strain, respectively (*Figure 2—figure supplement 1a and b*), along with the abundance of the corresponding top 5 molecules (VOCs or proteins) for each of the six sex/strain combinations (*Figure 2—figure supplement 1c–f*). Notably, major urinary protein 20 (Mup20/darcin) (*Roberts et al., 2010*), fatty acid-binding protein 5 (Fabp5) (*Furuhashi and Hotamisligil, 2008*), and N-acetylgalactosamine-6-sulfatase (Galns) (*Hori et al., 1995*) exhibit substantial power to discriminate sex across strains (*Figure 2—figure supplement 1b and f*). While Mup20 and Galns are considerably more abundant in male urine, Fabp5 appears to be specific for female samples. Strain discrimination, on the other hand, is optimal with protein-tyrosine kinase 2-beta (Ptk2b) (*Lev et al., 1995*), RNase T2 (Rnaset2) (*Thorn et al., 2012*), prosaposin (Psap) (*O'Brien and Kishimoto, 1991*), lymphocyte antigen 6A-2/6E-1 (Ly6a) (*van de Rijn et al., 1989*), and superoxide dismutase 1 (Sod1) (*McCord and Fridovich, 1969*; *Figure 2—figure supplement 1b and d*). Specifically, Ptk2b is detected exclusively in C57BL/6 mice. Ly6a is absent in BALB/c animals, whereas Rnaset2 is largely missing in both C57BL/6 and wild mice. With proteins exhibiting stronger discriminative power than VOCs (*Figure 2c–h*), future studies will have to focus on these proteins to identify potential functions as vomeronasal chemosignals.

Overall, we have detected similar amounts of proteins/VOCs across strains. However, individual variability in protein and VOC content was significantly higher in wild mice than in both laboratory strains (*Figure 2—figure supplement 2a*). We confirmed secretion of previously reported putative semiochemicals (*Hurst et al., 2001*; *Kaur et al., 2014*; *Leinders-Zufall et al., 2000*; *Novotny, 2003*; *Roberts et al., 2012*), including both VOCs (*Figure 2—figure supplement 2b*) and proteins (*Figure 2—figure supplement 2c*) in both wild and laboratory mice. Notably, however, all Mups (including Mup20) and most such VOCs were found in samples from either sex, albeit at male-biased concentrations for Mup3, Mup17, and Mup20 (*Figure 2—figure supplement 2c*). The only compounds showing male-specific secretion are 2-sec-butyl-4,5-dihydrothiazole and farnesenes (*Figure 2—figure supplement 2b*), which have previously been implicated as facilitators of female mouse puberty acceleration (*Novotny, 2003*).

Notably, we find a rich repertoire of 27 lipocalins in mouse urine. When based exclusively on lipocalin content, hierarchical clustering groups individual samples into a set of clusters that, with very few exceptions (5 out of 60), correspond to the six sex/strain combinations (*Figure 2—figure supplement*

**Table 1.** List of volatile organic compounds (VOCs) that – according to sparse partial least-squares discriminant analysis (sPLS-DA) – display the most discriminative power to classify samples by sex (top; related to *Figure 2c*) or strain (bottom; related to *Figure 2d*). Note that for all VOCs that facilitate sex discrimination, x-variate 1 is most informative. VOCs that are potentially important for strain differentiation, however, are best separated by either x-variate 1 or 2. Entries list internal mass spectrometry identifiers (int ID), identifiers extracted from MS analysis database (MS-DB-ID), the sex or strain that drives separation (bias), which two-dimensional component/x-variate represents the most discriminative variable (comp), PubChem chemical formula (Chem-ID), PubChem common or alternative name (alt. name), Chemical Entities of Biological Interest (ChEBI) or PubChem Compound Identification (CID), and putative origin (ori).

| int ID | MS-DB-ID | bias/sex | comp | Chem-ID | alt. name | ChEBI / CID | ori |
|---|---|---|---|---|---|---|---|
| A140 | Methane, nitro- | Male | 1 | $CH_3NO_2$ | Nitrocarbol | 77701 | ? |
| A336 | Thiazole, 2-ethyl-4,5-dihydro- | Male | 1 | $C_5H_9NS$ | 2-Ethylthiazoline | 86896 | Rodents |
| A220 | Butanenitrile, 2-methyl- | Male | 1 | $C_5H_9N$ | 2-Cyanobutane | 51937 | Insects |
| A305 | 2-sec-Butylthiazole | Male | 1 | $C_7H_{11}NS$ | 2-(1-Methylpropyl)thiazole | 519539 | Mouse |
| A209 | 2-Butenal, 2-methyl- | Male | 1 | $C_5H_8O$ | Tiglic aldehyde | 88419 | ammals |
| A278 | Furan, 2,3-dihydro-4-(1-methylethyl)- | Male | 1 | $C_7H_{12}O$ | Furan | 35559 | Mammals |
| A102 | 2-Butanone | Female | 1 | $C_4H_8O$ | Butan-2-one | 28398 | Mammals |
| A497 | Benzyl alcohol | Female | 1 | $C_7H_8O$ | Phenylmethanol | 17987 | Mammals |
| A28 | Acetaldehyde | Female | 1 | $C_2H_4O$ | Ethanal | 15343 | Mammals |
| A248 | Butyric acid, 3-tridecyl ester | Female | 1 | $C_{17}H_{34}O_2$ | Tridecyl butyrate | 169401 | Rodents |
| A307 | 1-Octen-3-ol | Female | 1 | $C_8H_{16}O$ | 1-Vinylhexanol | 34118 | Ubiquitous |
| A765 | 1,2-Benzenedicarboxylic acid, butyl 2-methylpropyl ester | Female | 1 | $C_{16}H_{22}O_4$ | Butyl isobutyl phthalate | 519539 | Mammals |
| int ID | MS-DB-ID | bias/strain | comp | Chem-ID | alt. name | ChEBI/CID | ori |
| A51 | Trichloromethane | C57BL/6 | 2 | $CHCl_3$ | Chloroform | 6212 | Ubiquitous |
| A14 | Cyclopropane, ethyl- | C57BL/6 | 2 | $C_5H_{10}$ | Ethylcyclopropane | 70933 | Ubiquitous |
| A74 | Ethyl acetate | C57BL/6 | 2 | $C_4H_8O_2$ | Acetoxyethane | 8857 | Yeast |
| A10 | Ethylenimine | C57BL/6 | 2 | $C_2H_5N$ | Aziridine | 9033 | Ubiquitous |
| A123 | n-Propyl acetate | Wild | 2 | $CH_3COOCH_2CH_2CH_3$ | Propyl ethanoate | 40116 | Insects |
| A355 | Pyrazine, 2-ethenyl-6-methyl- | Wild | 2 | $C_7H_8N_2$ | Pyrazine, 2-methyl-6-vinyl- | 518838 | Ubiquitous |
| A659 | Dodecanoic acid, isooctyl ester | Wild | 2 | $C_{20}H_{40}O_2$ | Isooctyl laurate | 2216918 | Plants |
| A165 | 2-Pentanone | Wild | 2 | $C_5H_{10}O$ | Ethyl acetone | 7895 | Ubiquitous |
| A372 | Pyridine, 2,3,4,5-tetrahydro- | Wild | 1 | $C_5H_9N$ | Piperideine | 47858 | Ubiquitous |
| A486 | Benzenamine, 3-methyl- | Wild | 1 | $C_6H_4CH_3NH_2$ | 3-Toluidine | 7934 | ? |
| A156 | Pentanal | BALB/c | 2 | $C_5H_{10}O$ | Valeraldehyde | 8063 | Insects |
| A181 | Isopropylsulfonyl chloride | BALB/c | 2 | $C_3H_7ClO_2S$ | 2-Propanesulfonyl chloride | 82408 | Bacteria |
| A456 | Cyclohexane, isothiocyanato- | BALB/c | 2 | $C_7H_{11}NS$ | Isothiocyanocyclohexane | 14289 | ? |
| A45 | Heptane, 2,4-dimethyl- | BALB/c | 2 | $C_9H_{20}$ | 2,4-Dimethylheptane | 16656 | Insects |
| A265 | Benzene, 1,2,3-trimethyl- | BALB/c | 1 | $C_9H_{12}$ | Hemellitol | 1797279 | Insects |
| A251 | Benzene, 1,2,4-trimethyl- | BALB/c | 1 | $C_9H_{12}$ | Pseudocumol | 7247 | Insects |
| A977 | Propene | BALB/c | 1 | $CH_2CHCH_3$ | Methylethylene | 6378 | ? |

**Table 2.** List of volatile organic compounds (VOCs) that – according to Random Forest classification and resulting Gini importance scores – display the most discriminative power to classify samples by sex (top; related to **Figure 2—figure supplement 1a**) or strain (bottom; related to **Figure 2—figure supplement 1b**).

As for several compounds listed in *Table 1*, many VOCs that display high relative rankings of individual variable relevance are common metabolites. Generally, it is reassuring that several VOCs are listed in both *Table 1* and Table 2, emphasizing that two different supervised machine learning algorithms (i.e. sPLS-DA [*Table 1*] and Random Forest [Table 2]) yield largely congruent results. Here, entries (in blue if identified by both sPLS-DA and Random Forest) list internal mass spectrometry identifiers (int ID), identifiers extracted from MS analysis database (MS-DB-ID), the sex or strain that drives separation (bias), PubChem chemical formula (Chem-ID), PubChem common or alternative name (alt. name), Chemical Entities of Biological Interest (ChEBI) or PubChem Compound Identification (CID), and putative origin (ori).

| nt ID | MS-DB-ID | bias/sex | Chem-ID | alt. name | ChEBI / CID | ori |
|-------|----------|----------|---------|-----------|-------------|-----|
| A140 | Methane, nitro- | Male | $CH_3NO_2$ | Nitrocarbol | 77701 | ? |
| A220 | Butanenitrile, 2-methyl- | Male | $C_5H_9N$ | 2-Cyanobutane | 51937 | Insects |
| A235 | 3-Penten-2-one | Male | $C_5H_8O$ | (E)-pent-3-en-2-one | 637920 | Insects |
| A336 | Thiazole, 2-ethyl-4,5-dihydro- | Male | $C_5H_9NS$ | 2-Ethylthiazoline | 86896 | Rat |
| A13 | Methanethiol | Male | $CH_3SH$ | MTMT | 878 | Mammals |
| A20 | Methylamine, *N,N*-dimethyl- | Male | $C_3H_9N$ | Trimethylamine | 1146 | Mouse |
| A765 | 1,2-Benzenedicarboxylic acid, butyl 2-methylpropyl ester | Female | $C_{16}H_{22}O_4$ | Butyl isobutyl phthalate | 519539 | Mammals |
| A240 | 2-Penten-1-ol, (Z)- | Male | $C_5H_{10}O$ | trans-2-Pentenol | 5364920 | Insects |
| A209 | 2-Butenal, 2-methyl- | Male | $C_5H_8O$ | Tiglic aldehyde | 88419, CID:5321950 | Mammals |
| A305 | 2-sec-Butylthiazole | Male | $C_7H_{11}NS$ | 2-(1-Methylpropyl)thiazole | 519539 | Mouse |
| A337 | 2-sec-Butylthiazole (2nd variant) | Male | $C_7H_{11}NS$ | 2-(1-Methylpropyl)thiazole | 519539 | Mouse |
| A102 | 2-Butanone | Female | $C_4H_8O$ | Butan-2-one | 28398 | Mammals |
| A152 | 2,4-Dimethyl-1-hexene | Male | $C_8H_{16}$ | 2,4-Dimethylhex-1-ene | 519301 | ? |
| A618 | 6-Methyl-2-pyridinecarbaldehyde | Male | $C_7H_7NO$ | 6-Methylpicolinaldehyde | 70737 | ? |
| A84 | 4-Hexen-3-one, 5-methyl- | Male | $C_7H_{12}O$ | 5-Methylhex-4-en-3-one | 256081 | Insects |
| int ID | MS-DB-ID | bias/strain | Chem-ID | alt. name | ChEBI / CID | ori |
| A51 | Trichloromethane | C57BL/6 | $CHCl_3$ | Chloroform | 6212 | Ubiquitous |
| A265 | Benzene, 1,2,3-trimethyl- | BALB/c | $C_9H_{12}$ | Hemellitol | 1797279 | Insects |
| A63 | Acetone | C57BL/6 | $C_3H_6O$ | Dimethyl ketone | 180 | Ubiquitous |
| A14 | Cyclopropane, ethyl- | C57BL/6 | $C_5H_{10}$ | Ethylcyclopropane | 70933 | Ubiquitous |
| A172 | (S)-(+)-2-Pentanol | Wild | $C_5H_{12}O$ | 2-Pentanol | 22386 | Mouse |
| A222 | 2-Hexanone | Wild | $C_6H_{12}O$ | Hexanone | 11583 | Mouse |
| A165 | 2-Pentanone | Wild | $C_5H_{10}O$ | Ethyl acetone | 7895 | Ubiquitous |
| A251 | Benzene, 1,2,4-trimethyl- | BALB/c | C9H12 | Pseudocumol | 7247 | Insects |
| A123 | *n*-Propyl acetate | Wild | $C_5H_{10}O_2$ | Propyl acetate | 7997 | Insects |
| A355 | Pyrazine, 2-ethenyl-6-methyl- | Wild | $C_7H_8N_2$ | Pyrazine, 2-methyl-6-vinyl- | 518838 | Mammals |
| A74 | Ethyl acetate | C57BL/6 | $C_4H_8O_2$ | Acetoxyethane | 8857 | Yeast |
| A156 | Pentanal | BALB/c | $C_5H_{10}O$ | Valeraldehyde | 8063 | Insects |
| A146 | Cyclohexene,3-(1-methylpropyl)- | Wild | $C_{10}H_{18}$ | 3-Sec-butyl-1-cyclohexene | 139917 | Insects |
| A194 | o-Xylene | Wild | $C_6H_4(CH_3)_2$ | Ortho-Xylene | 7237 | ? |
| A83 | Benzene | Wild | $C_6H_6$ | Benzole | 241 | Insects |

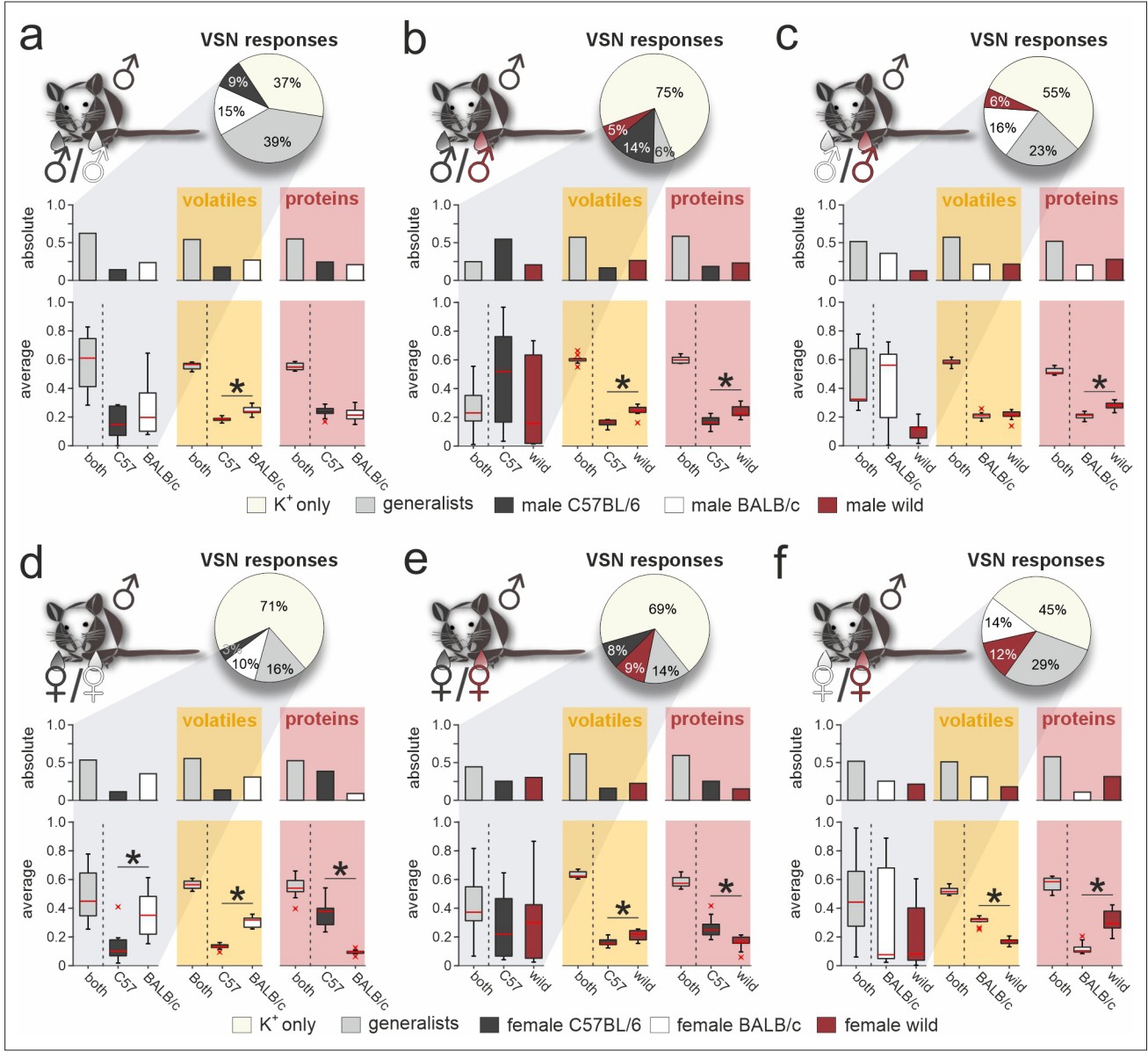

**Figure 3.** Selective vomeronasal sensory neuron (VSN) response profiles upon stimulation with strain-specific signatures. Quantitative comparison between VSN responses to paired stimuli and their respective chemical signatures. Neurons of male C57BL/6 mice were challenged with male (a–c) or female (d–f) urine, respectively. Response profiles are compared upon stimulation with C57BL/6 *versus* BALB/c urine (a and d), C57BL/6 *versus* wild stimuli (b and e), and BALB/c *versus* wild urine (c and f), respectively. Pie charts (top) illustrate the proportions of generalist (light gray) and specialist neurons (dark gray, white, and purple, respectively) among all K+-sensitive VSNs (a, n=1855; b, n=4116; c, n=2462; d, n=2376; e, n=3230; f, n=3377). Bar graphs (left) break down the urine-sensitive neurons by categories and compare their distribution with the proportions of volatile organic compounds (VOCs) (middle; yellow background; a, n=450; b, n=448; c, n=479; d, n=480; e, n=405; f, n=492) and proteins (right; red background; a, n=317; b, n=330; c, n=334; d, n=657; e, n=715; f, n=584) found either in both urine types (gray bars) or exclusively in samples from one group (color code as in pie charts). Box-and-whisker plots (bottom) illustrate generalist-to-specialist VSN category ratios over individual experiments (a, n=6; b, n=10; c, n=7; d, n=9; e, n=10; f, n=12). Boxes represent the first-to-third quartiles. Whiskers represent the 10th and 90th percentiles, respectively. Outliers (1.5 IQR; red x) are plotted individually. The central red band represents the population median (P$_{0.5}$). Results are shown in relation to box-and-whisker plots that outline chemical content data obtained from paired comparisons (n=10 individuals per group). Asterisks (*) indicate statistical significance, p<0.05, Wilcoxon signed-rank test.

*3*). This finding, therefore, demonstrates the power of urinary lipocalins for potential chemosensory discrimination of sex and/or strain.

## VSNs are more selective for strain than for sex

Next, we challenged neurons with stimulus pairs from two same-sex/different-strain combinations and asked whether VSN response profiles reflect the molecular fingerprint of corresponding urine samples (e.g. male C57BL/6 *versus* male BALB/c; *Figure 3a*). We analyzed a total of 17,416 K⁺-sensitive VSNs (*Figure 3a–f*). Again, we distinguished between specialist VSNs that responded exclusively to one stimulus, and generalists, which responded to both stimuli. Along the same lines, we categorized either strain-specific or broadly detected VOCs and proteins. Several conclusions emerge from these classifications: (i) With one exception (i.e. upon stimulation with male C57BL/6 *versus* wild stimuli; *Figure 3b*), roughly half of all urine-sensitive VSNs are generalists – a result consistent with our finding that generally ~50% of urine molecules are shared among compared strains. Accordingly, the fraction of strain-selective (specialist) responses is considerably larger than observed for sex-specific responses (*Figure 1e*). (ii) In female urine, BALB/c-specific proteins are substantially underrepresented, a fact not reflected by VSN response profiles (*Figure 3d and f*). (iii) Surprisingly, the amount of strain-specific molecules in wild mouse urine does not vastly exceed that in inbred strains; and (iv) accordingly, selective VSN responses to wild stimuli are by no means more common (*Figure 3b, c, e, and f*).

## Pronounced strain-dependent concentration imbalances between common urinary compounds are not reflected by generalist VSNs

When comparing generalist VSN responses to male *versus* female C57BL/6 urine (*Figure 1e and h*), we noted that the narrow normally distributed stimulus response index histograms did not match the broader and heterogeneous distributions of concentration disparities between sexes (*Figure 1h*; *Figure 1—figure supplement 1b-d*). For sex-dependent cues, as mentioned above, this could indicate that semiochemical concentration differences carry only limited information. Next, we therefore asked whether strain-specific concentration differences between urine samples exist and, if so, whether such differences can convey information about strain. In total, we recorded strain-independent generalist responses from 3366 VSNs in male C57BL/6 mice (*Figure 4*). In all but one experimental condition (i.e. when comparing male stimuli from BALB/c *versus* wild mice), response preference indices were normally distributed and well fit by relatively narrow Gaussian curves that centered around zero. By contrast, chemical analysis revealed that many compounds identified in urine from both strains differ substantially in concentration. For proteins, in particular, strong concentration disparities exist in all strain combinations analyzed. In both male and female samples, concentrations of common proteins are relatively low in C57BL/6, moderate in BALB/c, and high in wild animals. As shown in *Figures 2b and 3*, this concentration bias toward wild proteins does not translate into any dramatic overrepresentation of proteins selectively found in wild male and/or female urine. In fact, the massively skewed distributions of protein concentration indices are not reflected by generalist VSN profiles. The latter better match VOC concentration distributions, which generally display broad, yet Gaussian shapes. We conclude that VSN population response strength might not be so strongly affected by strain-dependent concentration differences among common urinary proteins. In that case, it would appear somewhat unlikely that individual VSN activity provides fine-tuned information about distinct semiochemical concentrations. Alternatively, as some (or even many) of the identified proteins could not serve as vomeronasal ligands at all, generalist VSNs might sample information from only a subset of compounds which, in fact, are secreted at roughly similar concentrations.

## Vomeronasal representation of female semiochemicals differs between two inbred strains

So far, our approach was restricted to VSN signals recorded from male C57BL/6 mice. An important question, of course, is whether the response profiles we observed are themselves recipient strain-dependent. Thus, we next aimed to assess the extent to which our findings generalize to VSN populations from another laboratory animal strain. We therefore repeated all six pairwise same-sex/different-strain stimulation experiments, using acute VNO slices from male BALB/c mice (*Figure 5* and *Figure 5—figure supplement 1*). Categorization as generalist or specialist VSNs revealed that proportions differed significantly between VSNs from BALB/c *versus* C57BL/6 mice, albeit at varying

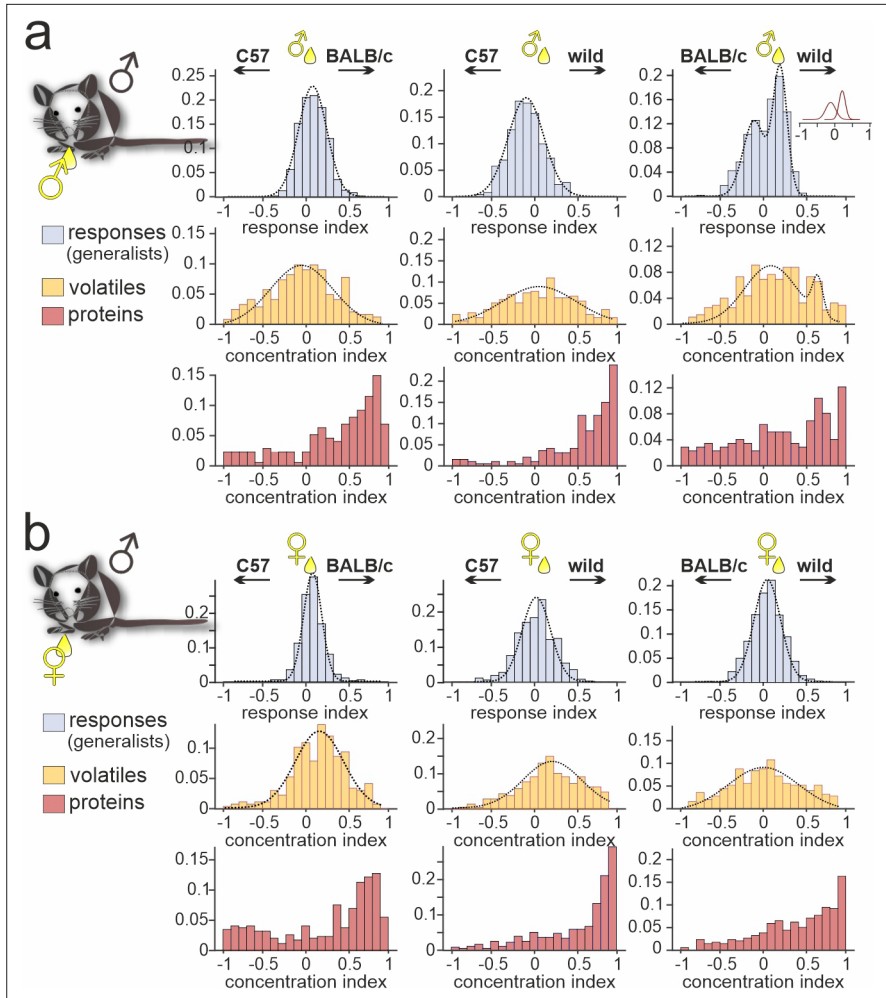

**Figure 4.** Strain-dependent concentration imbalances exert relatively mild effects on vomeronasal sensory neuron (VSN) population response homogeneity. Comparison of male C57BL/6 generalist VSN response preferences, upon stimulation with paired urine stimuli from different strains, with strain-dependent concentration imbalances among volatile organic compounds (VOCs) and proteins, respectively. (**a, b**) Response index histograms (top rows) depict distributions of generalist data outlined in *Figure 3* (gray bars). With one exception (**a** (right), male BALB/c *versus* male wild), histograms are well fitted by single Gaussian curves (dashed lines) that each center relatively close to zero (**a** (left), peak = 0.08, σ=0.18; **a** (middle), peak = –0.11, σ=0.21; **a** (right), 1st peak = –0.12, 1st σ=0.14; 2nd peak = 0.21, 2nd σ=0.09; **b** (left), peak = 0.07, σ=0.12; **b** (middle), peak = 0.03, σ=0.23; **b** (right), peak = 0.05, σ=0.18). Concentration index histograms (middle and bottom rows), calculated for VOCs (yellow) and proteins (red) found in both tested urine samples, are more heterogeneous. Notably, while most VOC concentration index histograms are also fitted by single, albeit broader Gaussian curves (**a** (left), peak = –0.07, σ=0.37; **a** (middle), peak = 0.02, σ=0.45; **a** (right), 1st peak = 0.07, 1st σ=0.32; 2nd peak = 0.64, 2nd σ=0.06; **b** (left), peak = 0.15, σ=0.29; **b** (middle), peak = 0.21, σ=0.35; **b** (right), peak = –0.01, σ=0.42), protein concentration imbalances are not normally distributed.

degrees. Similar to C57BL/6 neurons, selectivity of BALB/c neurons to wild-derived stimuli was rather rare. In fact, (i) compared to an average specialist rate of 11.2% ± 6.6% (mean ± SD) calculated over all 13 binary stimulus pairs (n=26 specialist types), we observed only few specialist responses upon stimulation with urine from wild females (2% and 3%, respectively), and (ii) we found comparatively few generalist signals when wild-derived female urine was among the paired stimuli (*Figure 5e and f*). This striking insensitivity is not observed in VSNs from C57BL/6 mice, suggesting substantial differences in VR expression between the two inbred laboratory strains. Notably, we can rule out that the observed differences result from differential sampling along the VNO's anterior-to-posterior axis since, for each

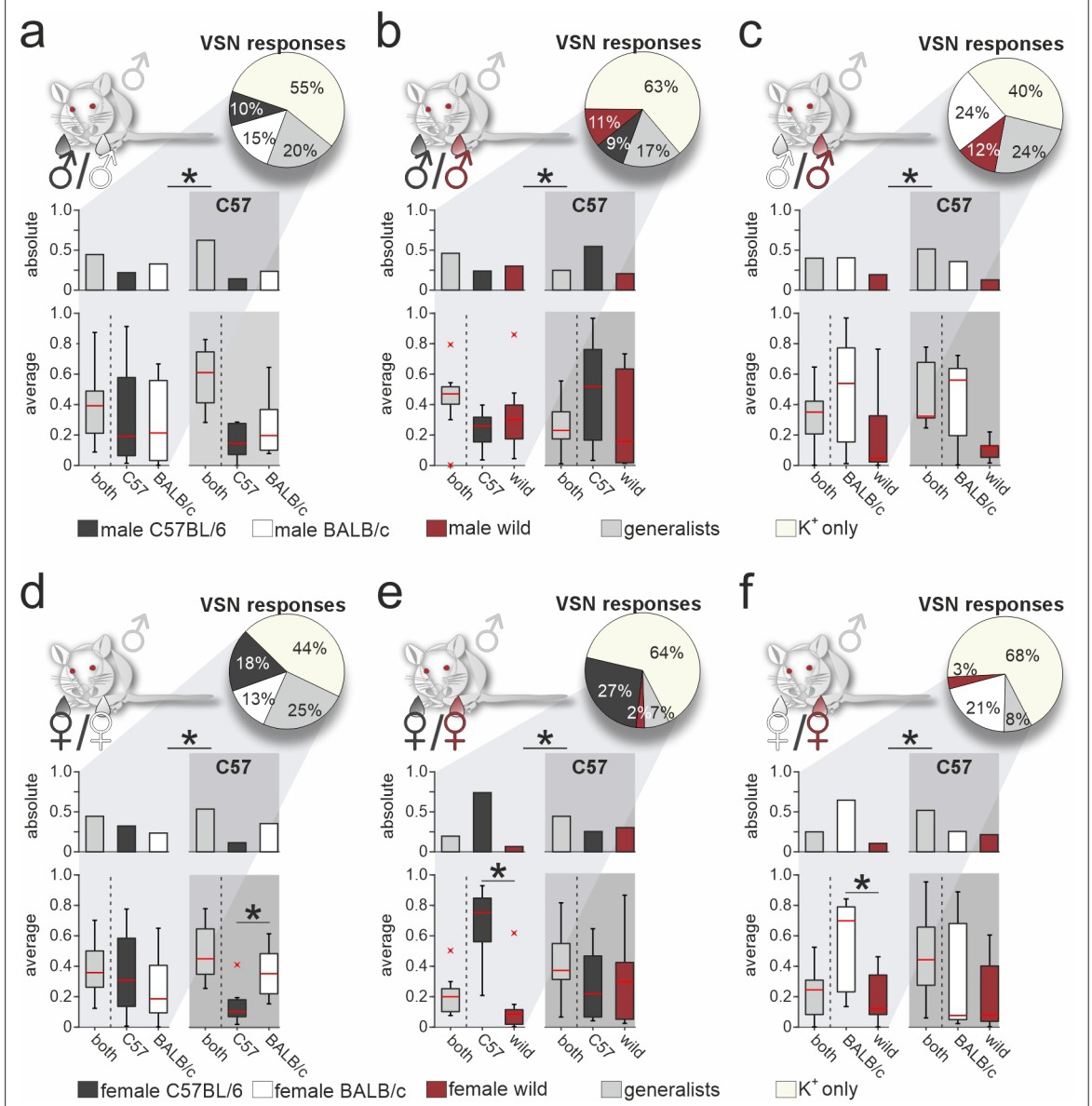

**Figure 5.** Vomeronasal representation of female semiochemicals differs between inbred strains. Comparison of vomeronasal sensory neuron (VSN) response profiles between male BALB/c and C57BL/6 mice. Pie charts (top) illustrate the proportions of generalist (light gray) and specialist BALB/c neurons (dark gray, white, and purple, respectively) among all K$^+$-sensitive VSNs (**a**, n=1244; **b**, n=2063; **c**, n=1903; **d**, n=1770; **e**, n=1934; **f**, n=1818). VSNs were challenged with male (**a–c**) or female (**d–f**) urine, respectively. Response profiles are compared upon exposure to C57BL/6 *versus* BALB/c urine (**a, d**), C57BL/6 *versus* wild stimuli (**b, e**), and BALB/c *versus* wild urine (**c, f**), respectively. Bar graphs break down the urine-sensitive neurons by categories and compare distributions among BALB/c neurons (left) to responses recorded from C57BL/6 VSNs (right; gray background). Box-and-whisker plots (bottom) illustrate generalist-to-specialist VSN category ratios over individual experiments (**a**, n=8; **b**, n=10; **c**, n=9; **d**, n=10; **e**, n=9; **f**, n=10). Boxes represent the first-to-third quartiles. Whiskers represent the 10th and 90th percentiles, respectively. Outliers (1.5 IQR; red x) are plotted individually. The central red band represents the population median (P$_{0.5}$). Asterisks (*) indicate statistical significance, p<0.05, Wilcoxon signed-rank test.

The online version of this article includes the following figure supplement(s) for figure 5:

**Figure supplement 1.** Comparison of generalist VSN response profiles between male BALB/c and C57BL/6 animals.

experiment, we routinely sample and average across slices cut along the organ's entire anterior-to-posterior length (see Materials and methods).

Next, we examined whether generalist VSN response profiles differ between male BALB/c and C57BL/6 animals (*Figure 5—figure supplement 1*). In total, we recorded generalist responses from 1741 BALB/c neurons. When plotting response preference indices, we noticed less homogeneous

distributions than previously observed in C57BL/6 mice. Five of the six histograms showed multiple peaks and thus could not be fitted by a single Gaussian, whereas the only histogram adhering to a normal distribution was comparatively broad (*Figure 5—figure supplement 1a*). As observed in C57BL/6 neurons, the skewed distributions of protein concentration indices were not reflected by BALB/c generalist VSN profiles. Comparison of generalist VSN response histograms between BALB/c and C57BL/6 mice (*Figure 5—figure supplement 1c and d*) revealed strong and consistent differences upon stimulation with female stimuli (*Figure 5—figure supplement 1d*). While single Gaussians around zero characterize C57BL/6 generalist distributions upon stimulation with female stimuli (*Figure 5—figure supplement 1d*), several prominent peaks emerged when fitting histograms derived from BALB/c VSNs. Notably, for some generalist BALB/c neurons, wild-derived female stimuli are less potent than their inbred strain counterparts. This finding either indicates reduced concentrations of the corresponding molecules in wild urine (rendering most proteins unlikely candidates; *Figure 5—figure supplement 1a and b*), or suggests some yet to be determined form of cooperativity upon receptor-ligand interaction. Together, for both generalist and specialist VSNs, vomeronasal representation of female semiochemicals differs considerably between the two inbred mouse strains.

In the present study, we have established a robust VSN activity assay that allows pairwise comparison of neural selectivity and response strength upon stimulation with chemically defined natural stimuli. In-depth chemical analysis of sex- and strain-specific individual urine samples revealed that (i) large fractions of urine content are shared among mice of all sex/strain combinations; (ii) the amount of molecules selectively found in wild mouse urine does not dramatically exceed the urinary secretome of inbred strains; (iii) across strains, female-specific proteins vastly outnumber the male-specific variety, while (iv) overall protein concentration is substantially enriched in male urine; (v) concentrations of common proteins are relatively low in C57BL/6, moderate in BALB/c, and high in wild animals; (vi) both secreted VOC and protein profiles provide sufficient information to distinguish sex, strain, or both; and (vii) the rich urinary lipocalin repertoire alone might allow chemosensory discrimination of sex and/or strain.

When asking how much of this chemical information is accessible to inbred male mice via vomeronasal sampling, we observe that (i) VSN population response profiles do not reflect the global molecular content of urine, suggesting that the VNO functions as a rather selective molecular detector; (ii) selective VSN responses to wild stimuli are by no means more common (in fact, selectivity to wild-derived stimuli is rather rare); (iii) VSN generalist signal strength is unlikely to encode semiochemical concentrations across the entire range of compounds; (iv) male BALB/c neurons display striking insensitivity when challenged with urine from wild females; and, thus, (v) vomeronasal representation of female semiochemicals differs considerably between inbred strains.

## Discussion

Urine is the primary source of social chemosignals among mice (and, in fact, many other mammals) and contains both 'fixed' (i.e. genomic) information about strain, sex, individual identity, genetic histocompatibility and background, as well as 'variable' (i.e. metabolic) information on current social, reproductive, and health status (*Hurst, 2005*). The ability to glean ethologically meaningful information from chemosensory sampling of urine (or any conspecific bodily secretion) depends on (i) the specific (semio)chemical composition of a urine sample, and (ii) the sensory apparatus used for sampling. For most mammals, the VNO is the key chemosensory structure involved in detecting conspecific chemical cues (*Mohrhardt et al., 2018*). The virtually universal use of inbred laboratory mice in research aimed at understanding VNO physiology – as both the donors of stimuli and the experimental subjects employed in these studies – could have resulted in misconceptions and biased notions about vomeronasal signaling and, thus, conspecific chemical communication. If that were the case, one might question the relevance and ethological validity of conclusions drawn from a large body of work using inbred secretions (*Bansal et al., 2021*). Here, we addressed this issue from both a chemical ecology and a physiological perspective. In-depth comparative molecular profiling of urine from two classical laboratory strains as well as wild mice reveals several shared features, but also qualitative and quantitative differences in composition. Furthermore, we observe substantial differences in vomeronasal representations of stimuli between C57BL/6 and BALB/c sensory neuron populations.

For analytical purposes, we separate the urinary 'volatilome' and proteome. In chemosensory research, this distinction has often been conceptualized as general (i.e. airborne) odors, which activate

the main olfactory system, *versus* vomeronasal stimuli (*Mucignat-Caretta et al., 2012*). This notion, however, is misleading since organic compounds with low molecular weight and high vapor pressure (i.e. VOCs) in bodily secretions do not instantly evaporate, of course. Rather, they are readily accessible for vomeronasal sampling upon direct contact during investigatory behavior. Notably, our chemical profiling approach omits (sulfated) steroids, other non-volatile small organic molecules, which have previously been identified in mouse urine as VSN stimuli (*Nodari et al., 2008*). Caution should thus be exerted to not attempt to fully explain VSN response specificity based on VOC and protein content alone.

For both urinary VOCs and proteins, large fractions are shared among both male and female mice of either genetic background (*Figure 2a and b*). Such compounds could be considered generic (mouse) urine components and might not even serve any chemosensory signaling functions. Notably, both the urinary volatilome and proteome on their own, each entail sufficient information to discern an individual's sex and strain, with protein content exhibiting stronger discriminative power. The protein that is most informative for discriminating between sexes is, perhaps not surprisingly, Mup20 (darcin) (*Roberts et al., 2010*). This well-described 'maleness signal' had previously been reported to elicit innate attraction and generate a conditioned place preference in females (*Demir et al., 2020*; *Roberts et al., 2012*), whereas, in males, Mup20 promotes aggression (*Kaur et al., 2014*). Chemosensory roles of the second and third best protein determinants of sex discrimination are basically unexplored. Fabp5 (*Furuhashi and Hotamisligil, 2008*) and Galns (*Hori et al., 1995*) are substantially enriched in female and male urine, respectively. Fatty acid-binding proteins, including Fabp5, are evolutionary conserved intracellular lipid chaperones that coordinate cellular lipid trafficking and signaling and are thus linked to metabolic and inflammatory pathways (*Furuhashi and Hotamisligil, 2008*). *N*-Acetylgalactosamine-6-sulfatase (Galns) on the other hand is a lysosomal hydrolase. Five urinary proteins – Ptk2b (*Lev et al., 1995*), Rnaset2 (*Thorn et al., 2012*), Psap (*O'Brien and Kishimoto, 1991*), Ly6a (*van de Rijn et al., 1989*), and Sod1 (*McCord and Fridovich, 1969*) – display pronounced strain-dependent differences in concentration. None of these have previously been attributed a chemosensory function. Challenging the mouse VNO with purified recombinant protein(s) will help elucidate whether such functions exist.

Proteomic profiling revealed three additional, rather unexpected findings: First, while male urine contains much higher protein concentrations, females of a given strain secrete a larger variety of proteins (including MUPs). In fact, we do not find a single protein that is exclusively detected in males across strains. By contrast, 23 urinary proteins, while present in all three strains, are found only in females. In line with these observations, only one additional Mup (i.e. Mup21) made the list of the 20 most informative proteins that discriminate sex. In general, our data provide little evidence for a sparse molecular code of (fe)maleness. Rather, the concept of 'signature mixtures' (*Ben-Shaul, 2015*; *Kahan and Ben-Shaul, 2016*; *Stopková et al., 2023*; *Wyatt, 2017*), which emphasizes a combinatorial ratio code instead of mere presence/absence phenomena, gains traction. Second, we identify a surprisingly rich lipocalin repertoire in urine, which alone could allow chemosensory discrimination of sex/strain combinations. Within a total of 27 lipocalins, individual patterns allow hierarchical clustering into sex/strain-specific groups (*Figure 2—figure supplement 3*). These data, thus, support the notion that the lipocalin 'code', if relevant, is combinatorial. Third, regarding its molecular spectrum, the urinary 'secretome' in wild mice does not differ dramatically from the repertoire found in laboratory strains. To paraphrase this generally reassuring conclusion: while inbreeding could have dramatically modified the nature of chemical secretions and, consequently, their perception by other mice, inbred chemostimuli are largely representative of (potentially) ethologically more relevant wild chemosignals. Notably, however, while mean VOC and protein concentrations show similar distributions across sex/strain combinations, individual variability is strongly increased in wild mouse urine (*Figure 2—figure supplement 2a*). Accordingly, this finding confirms previous reports of increased individual variation in wild mice (*Beynon et al., 2002*; *Cheetham et al., 2009*). Another inherent factor that could account for differences in urine secretions among each of the groups, and particularly for comparisons between inbred and wild stimuli, is the microbiome (*Bansal et al., 2021*; *Moudra et al., 2021*). Yet, we stress that all urine donors were housed in the same facility and fed with the same diet (*Bansal et al., 2021*).

Some limitations of our study need to be acknowledged. Regarding stimuli, the six secretion sets we used do not cover the entire coding capacity of the accessory olfactory system (*Isogai et al., 2011*). Moreover, for VSN response profiling, stimulus samples were pooled across 10 individuals in

each of the 6 sex/strain categories. While pooling stimuli reduces individual variability across samples (e.g. regarding fluctuating physiological states), relevant stimulus aspects could be masked. Thus, given the increased chemical variability we observed among individual wild urine samples, pooling might obscure distinctive molecular features of wild mouse secretions. The same holds true for estrus cycle-dependent female stimuli. Because we did not monitor the estrus stage of female urine donors, mixes likely contain samples across the entire cycle. This is relevant as VSN responses may be affected by a donor's cycle stage (*Bansal et al., 2021*; *Cichy et al., 2015*; *Uchida et al., 2014*). Another limitation stems from the use of male inbred mice as experimental subjects. The rationale behind this experimental strategy is to allow for comparisons with our previous study on AOB response profiles (*Bansal et al., 2021*), which used the exact same settings (see below). Nonetheless, future efforts will have to reveal (i) whether our main findings apply to female recipients; and (ii) if our observations generalize to other inbred, outbred, or wild mouse strains. As we recently declared (*Bansal et al., 2021*), the latter considerations underscore the importance of recording from wild recipients. While this endeavor presents significant practical challenges, it remains an important goal for future studies.

A surprising, but overall reassuring observation is that responses to wild and inbred stimuli are qualitatively similar. Strikingly, and somewhat counterintuitively, selective VSN responses to wild stimuli are rather rare. This does not result from a generally reduced compound content in wild urine as molecular profiling revealed comparably rich chemical portfolios in wild and inbred samples alike. Rather, we speculate that inbreeding over hundreds of generations in laboratory settings (*Phifer-Rixey and Nachman, 2015*) has resulted in 'microevolutionary' pressure to maintain sensitivity to signals from same- or similar-strain individuals. Indeed, compared to the C57BL/6 reference genome, genetic variability within VR repertoires is massively increased among wild-derived mice (particularly of the *M. musculus* subspecies) (*Wynn et al., 2012*). Among the approximately 200 orthologous receptor genes compared, BALB/c genes display 184 non-synonymous and just one private (i.e. unique to a given strain) single-nucleotide polymorphism (SNP). By contrast, VR genes of the wild-derived *M. musculus* PWK/PhJ strain show 789 non-synonymous SNPs and 508 private SNPs (*Wynn et al., 2012*).

Initially, we asked whether VSN response profiles reflect the global molecular content of urine or, by contrast, if the VNO serves as a rather selective semiochemical detector. Our findings support a high level of selectivity. When challenged with same-sex/different-strain stimuli, large VSN fractions selectively respond to just one stimulus. Intriguingly, this fraction of strain-selective specialists is considerably larger than observed for sex-specific responses. In line with the notion of highly selective vomeronasal sampling is our observation that the concentration differences between compounds shared among strains, which are often substantial, are not reflected by similarly pronounced differences in response strength among generalist VSNs. There are several, not necessarily mutually exclusive explanations for this finding: First, concentration could simply not be a read-out parameter for VSNs, which would support previous ideas of concentration-invariant VSN activity (*Leinders-Zufall et al., 2000*). Second, the concentrations in freshly released urine could just exceed the dynamic tuning range of VSNs since, particularly for VOCs, natural signals (e.g. in scent marks) must be accessible to a recipient for a prolonged amount of time (sometimes days). A similar rationale could explain the increased protein concentrations in male urine, since male mice use scent marking to establish and maintain their territories and urinary lipocalins serve as long-lasting reservoirs of VOCs (*Hurst et al., 1998*). Third, generalist VSNs might sample information only from a select subset of urinary compounds, which, given their role as biologically relevant chemosignals, might be released at tightly controlled (and thus similar) concentrations. In fact, in the most extreme scenario, several compounds that do display substantial strain- and/or sex-specific differences in concentration might not act as chemosignals at all. Forth, to some extent, different response profiles could be attributed to non-volatile small organic molecules such as steroids (*Nodari et al., 2008*), which were beyond the focus of our chemical analysis.

While, compared to wild-derived mice, the genetic differences in VR repertoires between C57BL/6 and BALB/c animals appear rather modest (*Wynn et al., 2012*) (see above), vomeronasal representation of female semiochemicals differs considerably between both inbred strains. We conclude that, even in closely related inbred mice, strain-to-strain VR variation must be prominent, an idea supported by various reports of differences in genetic VR makeup across strains (*Lilue et al., 2018*; *Miller et al., 2020*; *Park et al., 2011*). With monoallelic VR expression and 184 described non-synonymous SNPs between C57BL/6 and BALB/c receptor genes, it is likely that even individuals of the same

strain express functionally different arrays of VSNs. Adding another layer of complexity, state- and experience-dependent changes in VSN sensitivity have recently been described (*Dey et al., 2015*; *Xu et al., 2016*).

By adopting the same experimental design (i.e. using the same sets of stimuli in male C57BL/6 *versus* BALB/c mice) in both this study on VSN response profiles and our previous analysis of sensory representations in the AOB (*Bansal et al., 2021*), we provide a unique comparative perspective on signal transformation along the initial processing nodes of the accessory olfactory pathway. We observe several differences in representations of ethologically relevant urine stimuli between the VNO and AOB. Notably, while stimulus representations across the two inbred recipient strains were very similar in AOB recordings (*Bansal et al., 2021*), we here observe clear differences in VSN activity, particular upon stimulation with female stimuli. Moreover, a substantial fraction of AOB neurons were selective to wild rather than inbred stimuli, whereas relatively few VSNs showed such selectivity. Consistent with the elaborate wiring patterns within the AOB, these observations imply the presence of non-trivial transformations between VSN and AOB representations. Understanding the exact nature, purpose, and neuronal substrates of these transformations remains an important topic for future studies.

## Materials and methods

### Animals

All animal procedures were approved by local authorities at RWTH Aachen University, were performed in accordance with local Animal Care and Use Committees' regulations, and in compliance with European Union legislation (Directive 2010/63/EU) and recommendations by the Federation of European Laboratory Animal Science Associations. C57BL/6 and BALB/c mice (Charles River Laboratories, Sulzfeld, Germany) were housed in groups of both sexes (room temperature [RT]; 12:12 hr light-dark cycle; food and water available ad libitum). All $Ca^{2+}$ imaging experiments used slices from young male adults.

Urine collection from two strains of inbred mice (C57BL/6NCrl and BALB/cAnNCr) as well as first-generation offspring of wild mice was performed at Charles University (Prague, Czech Republic) according to the institute's ethical committee guidelines. Inbred mice were purchased from pathogen-free facilities of the Institute of Molecular Genetics (Czech Academy of Sciences in Prague). Wild *M. musculus* mice were caught in house shelters and agricultural buildings near Prague (Czechia), transferred to the local animal facility at Charles University, and bred for one generation. 20 male (M) and 27 female (F) wild mice were caught at six different sites in the broader Prague area (i.e. Bohnice [50.13415N, 14.41421E; 2M+4F], Dolni Brezany [49.96321N, 14.4585E; 3M+4F], Hodkovice [49.97227N, 14.48039E; 5M+6F], Písnice [49.98988N, 14.46625E; 3M+6F], Lhota [49.95369N, 14.43087E; 1M+2F], and Zalepy [49.9532N, 14.40829E; 6M+5F]). 18 of the 27 wild females were caught pregnant. The remaining 9 females were mated with males caught at the same site and produced offspring within a month. All mice that served as urine donors were fed on the same diet. Food and water for all strains were provided ad libitum under stable conditions (13:11 hr light-dark cycle; 23°C).

### Chemicals and solutions

The following solutions (**S₁–S₆**) were used:

(**S₁**) 4-(2-Hydroxyethyl)piperazine-1-ethanesulfonic acid (HEPES) buffered extracellular solution containing (in mM) 145 NaCl, 5 KCl, 1 $CaCl_2$, 1 $MgCl_2$, 10 HEPES, pH 7.3 (adjusted with NaOH), 300 mOsm (adjusted with glucose).

(**S₂**) Oxygenated (95% $O_2$, 5% $CO_2$) extracellular solution containing (in mM) 125 NaCl, 25 $NaHCO_3$, 5 KCl, 1 $MgSO_4$, 1 $CaCl_2$, 5 *N*,*N*-bis(2-hydroxyethyl)-2-aminoethanesulfonic acid (BES); pH = 7.3; 300 mOsm (adjusted with glucose).

(**S₃**) Elevated extracellular $K^+$ solution containing (in mM) 100 NaCl, 50 KCl, 1 $CaCl_2$, 1 $MgSO_4$, 10 HEPES; pH = 7.3 (adjusted with NaOH); 300 mOsm (adjusted with glucose).

If not stated otherwise, chemicals were purchased from Sigma (Schnelldorf, Germany). Solutions and stimuli were applied from air pressure-driven reservoirs via an eight-in-one multibarrel 'perfusion pencil' (Science Products, Hofheim, Germany). Changes in focal superfusion (*Veitinger et al., 2011*) were software-controlled and synchronized with data acquisition by transistor-transistor logic input to

12 V DC solenoid valves using a TIB 14S digital output trigger interface (HEKA Elektronik, Lambrecht/Pfalz, Germany).

## Stimuli

For all three donor types (C57BL/6NCrl, BALB/cAnNCr, and wild mice), we collected fresh urine by gentle bladder massage from 10 adult male and female individuals, respectively (resulting in a total of 60 individual samples), starting at an age of 10 weeks. When selecting wild individuals (first-generation offspring) for urine collection, we ensured that all six capture sites (see above) were represented and that age-matched animals displayed similar weight (~17 g). To minimize concentration differences that might result from sample-to-sample volume (i.e. dilution) variability, we collected and pooled four to six samples from each individual over several days until each of the 60 animals had provided a total urine volume of >500 µl. Next, we measured general protein content for each sample (Bradford assay). Aliquots of 10 µl were subjected to GCxGC-MS and nLC-MS/MS (see below).

The remaining samples were divided into ready-to-use aliquots and stored at −86°C. Prior to experiments, aliquots were thawed and diluted 1:100 in $S_1$ (*Hagendorf et al., 2009*). For each of the six strain/sex-specific stimulus combinations, we created pools from all 10 individuals to minimize individual-to-individual variability. For both inbred and wild female mice, estrus stage was not determined. However, urine collection over several days and pooling across 10 individuals in each stimulus set is designed to reduce variability. Notably, the urine samples employed in this study are the same stimuli used previously to compare sensory representations of inbred and wild stimuli in the AOB of male C57BL/6 and BALB/c mice (*Bansal et al., 2021*).

## Two-dimensional gas chromatography-mass spectrometry

Urine VOCs were sampled with Headspace Solid Phase Micro Extraction (HS SPME) on fiber (DVB/CAR/PDMS gray; Supelco, USA) after 5 min incubation at 55°C. Next, VOCs were analyzed using two-dimensional gas chromatography with mass detection (Pegasus 4D, LECO Europe B.V., Geleen, The Netherlands) with a combination of mid-polar and non-polar separation columns (primary column: SLB-IL60 [30 m × 0.25 mm, Sigma-Aldrich, USA]; secondary column Rxi-5sil MS [1.4 m × 0.25 mm, Restek, Australia]). Parameters were set as follows: inlet temperature 270°C, splitless injection mode, constant He flow 1 ml min$^{-1}$, modulation time 4 s (hot pulse 0.6 s), modulation temperature offset with respect to secondary oven 15°C. Temperature program for primary oven: 50°C (1 min), increase to 320°C (10 °C min$^{-1}$), 320°C (3 min). +5°C temperature offset on secondary column. Transfer line temperature was held at 250°C. Mass detector was equipped with an electron ionization source and time-of-flight analyzer enabling unit mass resolution (scanned mass range was 30–500 *m/z*). The ion source chamber was held at 250°C. ChromaTOF v4.5 software (LECO Europe B.V.) was employed for instrument control and data processing. Selected compounds were identified by mass spectra library matching (NIST MS 2.2, USA). Compounds identified only in blanks (or that were highly abundant in blanks) were removed from analysis (e.g. silanes, siloxanes, propylphosphines, etc.).

## Protein digestion and nLC-MS/MS

Urine proteins were precipitated with cold acetone and centrifuged (14,000 × *g*; 10 min; 0°C), followed by re-suspension of dried pellets in digestion buffer (1% SDC, 100 mM TEAB; pH 8.5). Next, protein concentration in each lysate was determined (BCA assay kit; Fisher Scientific). We used tris(2-carboxyethyl)phosphine (TCEP; 5 mM; 60°C; 60 min) as reducing agent and *S*-methyl methanethiosulfonate (MMTS; 10 mM; 10 min; RT) to block free cysteines. After trypsin digestion (1 µg per sample; 37°C; overnight), peptides were desalted on a Michrom C18 column. We used reverse-phase nano-columns (EASY-Spray column, 50 cm × 75 µm ID, PepMap C18, 2 µm particles, 100 Å pore size) for high-resolution peptide separation. Eluting peptide cations were converted to gas-phase ions by electrospray ionization and analyzed on a Thermo Orbitrap Fusion (Q-OT-qIT; Thermo Fisher, Waltham, MA, USA) as previously described (*Černá et al., 2017*; *Stopkova et al., 2017*). LC-MS data were pre-processed with MaxQuant software (version 1.5.3.8) (*Voukali et al., 2021*). The false discovery rate was set to 1% for both proteins and peptides. We specified a minimum peptide length of seven amino acids. The Andromeda search engine was used for MS/MS spectra search against Uniprot *M. musculus* database (downloaded June 2015), containing 44,900 entries. From this database, all MUP and OBP sequences were removed and replaced by complete lists of MUPs (Ensembl) and OBPs

(*Stopková et al., 2021*), respectively. We also added some sequences from TrEMBL that were missing in Uniprot (e.g. KLKs, BPIs, SPINKs, SCGB/ABPs, and LCNs). Enzyme specificity was set as C-terminal to Arg and Lys, also allowing cleavage at proline bonds (*Rodriguez et al., 2008*) and a maximum of two missed cleavages. Quantifications were performed using label-free algorithms (*Cox et al., 2014*) with a combination of unique and razor peptides.

## VNO slice preparation

For confocal $Ca^{2+}$ imaging, acute coronal VNO slices were prepared as previously described (*Cichy et al., 2015*; *Wong et al., 2018*). Briefly, C57BL/6 and BALB/c mice were euthanized by brief exposure to a $CO_2$ atmosphere, cervical dislocation, and decapitation. The lower jaw and palate were rapidly removed. The VNO was dissected, embedded in 5% low-gelling temperature agarose (VWR International, Erlangen, Germany), placed in ice-cold oxygenated $S_2$, and coronal slices (150 µm) were cut on a VT1000S vibrating microtome (Leica Biosystems, Nussloch, Germany). For each experiment, we routinely prepared VNO slices along the organ's entire anterior-to-posterior axis (not including the most anterior tip, where the VNO lumen tapers into the vomeronasal duct, and the most posterior part, the lumen "twists" toward the ventral aspect and its volume decreases; *Hamacher et al., 2024*). This usually yields ~7 slices per individual experiment/session. Therefore, we routinely sample and average across the entire VNO anterior-to-posterior axis for each experiment. Slices were transferred to a submerged, chilled, and oxygenated storage chamber with circulating $S_2$ until use.

## $Ca^{2+}$ imaging

In vitro imaging of VSN activity in acute coronal VNO slices was performed as described (*Wong et al., 2018*). Briefly, for bulk loading, slices were incubated (90 min; 5°C) in circulating $S_2$ with the $Ca^{2+}$-sensitive dye CAL520/AM (4.5 µM; Biomol, Hamburg, Germany) and 0.05% Pluronic F-127 (20% solution in DMSO; Thermo Fisher Scientific, Schwerte, Germany). After washing (5×, $S_2$), slices were transferred to a recording chamber (Luigs & Neumann, Ratingen, Germany) mounted on an upright fixed-stage scanning confocal microscope (TCS SP5 DM6000CFS, Leica Microsystems) equipped with a 20×/1.0 NA water immersion objective (HCX APO L, Leica Microsystems), and infrared-optimized differential interference contrast optics. Slices were continuously superfused with oxygenated $S_2$ (~5 ml $min^{-1}$; gravity flow). CAL520 was excited at 488 nm (multi-line argon laser; <25% laser power) and fluorescence was detected within a 500–600 nm spectral band. Changes in cytosolic $Ca^{2+}$ were monitored over time at 1.0 Hz frame rate (1024×512 pixels; 400 Hz bidirectional scanning frequency) using LAS AF software (Leica Microsystems).

## Experimental design and statistical analysis

*$Ca^{2+}$ imaging in VNO slices* – All data were obtained from independent experiments performed in ≥3 sessions using ≥3 different animals. Individual numbers of cells/experiments (n) are denoted in figures and/or captions. Since, for each experiment, we prepare slices along the VNO's anterior-to-posterior axis (see above), we routinely sample and average across the entire VNO anterior-to-posterior axis for each experiment. Data were analyzed offline using Leica LAS AF 2.4 (Leica Microsystems), ImageJ 1.51n (Wayne Rasband, National Institutes of Health, USA), MATLAB R2017b (MathWorks, Natick, MA, USA), and Excel (Microsoft, Seattle, WA, USA) software. If not stated otherwise, results are presented as box-and-whisker plots, where boxes represent the first-to-third quartiles and whiskers represent the 10th and 90th percentiles, respectively. Outliers (1.5 IQR) are plotted individually. The central band represents the population median ($P_{0.5}$). Statistical analyses were performed using unpaired t-tests, Wilcoxon signed-rank tests, Mann-Whitney U tests, and two-sample Kolmogorov-Smirnov tests (as dictated by data distribution and experimental design). Tests and corresponding p-values that report statistical significance (≤0.05) are individually specified in figure captions.

For individual cell analysis, regions-of-interest were defined to outline essentially all depolarization-sensitive ($S_3$) somata per field of view, based on DIC imaging of cell morphology at rest. After movement correction (StackReg/Rigid Body transformation plugin [*Thévenaz et al., 1998*] in ImageJ) of time-lapse image stacks, changes in relative fluorescence intensity were calculated as ΔF/F and measured in arbitrary units. Neurons were classified as 'responsive' when showing stimulus-dependent $Ca^{2+}$ elevations in somata according to the following three criteria (*Fluegge et al., 2012*; *Wong et al., 2018*): (i) exposure to high extracellular $K^+$ concentrations (50 mM; $S_3$) induced a robust $Ca^{2+}$ transient; (ii) for

at least one stimulation with diluted urine, a transient increase in fluorescence intensity was observed during the stimulation period; and (iii) the signal peak intensity exceeded the average prestimulation baseline intensity plus two standard deviations for a continuous period of at least 3 s ($I_{resp} > I_{baseline} + 2 \times SD(I_{baseline})$). All responses were normalized to positive controls (i.e. responses evoked by elevated extracellular $K^+$). Responsive neurons were categorized as: (i) neurons only sensitive to depolarization ($K^+$); (ii) *specialist* neurons that selectively responded to one of two presented urine stimuli (and $K^+$); and (iii) *generalist* neurons that responded to both urine stimuli (and $K^+$). Raw data (i.e. original intensity *versus* time traces) from each responsive cell were visually inspected to control for potential unspecific signals (e.g. caused by spontaneous activity). Based on peak amplitudes recorded from each urine-sensitive neuron (both generalists and specialists), we calculated a cell-specific *response index* (RI):

$$RI = \frac{mean\left(\frac{\Delta F}{F}\right)_{stim1} - mean\left(\frac{\Delta F}{F}\right)_{stim2}}{mean\left(\frac{\Delta F}{F}\right)_{stim1} + mean\left(\frac{\Delta F}{F}\right)_{stim2}}$$

where mean ($^{\Delta F}/_F$) is the average signal amplitude evoked by consecutive stimulation with the same stimulus (i.e. either stim1 or stim2). As a measure of how reliably the same stimulus evokes a response upon consecutive exposures, we additionally calculated a cell-specific *reliability index* (ReI), which is based either on response amplitudes or their integrals (i.e. area under curve [AUC]):

$$ReI = \frac{response\left(\frac{\Delta F}{F} \vee AUC\right)_{1ststim} - response\left(\frac{\Delta F}{F} \vee AUC\right)_{2ndstim}}{response\left(\frac{\Delta F}{F} \vee AUC\right)_{1ststim} + response\left(\frac{\Delta F}{F} \vee AUC\right)_{2ndstim}}$$

where consecutive responses of similar strength (i.e. high reliability) are reflected by a small ReI value.

*Proteomics and metabolomics* – Pairwise comparison between different urine samples allowed comparative analysis of physiological activity (VSN $Ca^{2+}$ signals) *versus* urine composition (molecular content). Chemical content was categorized as follows: (i) a specific compound is considered 'present' in urine from a given sex/strain combination (i.e. male or female BALB/c, C57BL/6, or wild, respectively) if it is identified in ≥3 out of 10 individual samples; (ii) a compound is thus considered 'absent' if it is found only in ≤2 individual samples. Accordingly, compounds that are detected as 'present' in urine samples from both sex/strain combinations being compared are designated as *generic* in binary comparisons. By contrast, a compound that is 'present' in only one of the two sex/strain combinations is designated as *specific*. To identify compounds that are *enriched* in a given sex/strain combination (*Figure 2a and b*), we raised the criterion for a 'present' call to identification in ≥6 out of 10 individual samples.

To quantify concentration differences between samples, we calculated *concentration indices* (CIs) for generic compounds, both VOCs (*GCxGC-MS*) and peptides/proteins (*nLC-MS/MS*):

$$CI = \frac{meanconc.\left(X_{group1}\right)_- meanconc.\left(X_{group2}\right)}{meanconc.\left(X_{group1}\right)_+ meanconc.\left(X_{group2}\right)}$$

where mean conc. is the average concentration of a given compound X among all 10 samples within a group (i.e. specific sex/strain combination).

*Proteomic and metabolomic analysis* – Because both GCxGC-MS and nLC-MS/MS data have similar (negative binomial) distributions, we processed them using the same procedure. First, we performed sPLS-DA (*Rohart et al., 2017*) to detect potential sources of variation in quantile normalized datasets. Next, for pairwise comparisons, data reduction eliminated all table entries (rows) if a given metabolite or protein was detected only in ≤3 individuals in both groups. However, if a compound was found in samples from ≥4 individuals in a group (e.g. in males or females), we included the full table entry (row). Accordingly, a chemical is considered 'unique'/group-specific if found in ≥4 individual samples from one group, but is missing in all 10 samples of the other group. For calculation of sexual dimorphism, we used the power law global error model (*Pavelka et al., 2004*) to detect differentially expressed/abundant proteins and VOCs. To detect the importance of significant molecules in discriminating either sex

or strain, we used a machine learning technique – Random Forest for Classification (*Breiman, 2001*) (implemented in R software) – and inferred feature importance scores (Gini importance; *Breiman, 2001*) that provide relative rankings of individual variable relevance. nLC-MS/MS proteomics data have been deposited to the ProteomeXchange Consortium via the PRIDE (*Perez-Riverol et al., 2022*) partner repository with the dataset identifier PXD042324. Metabolomics data have been deposited to the EMBL-EBI MetaboLights database with the identifier MTBLS7439.

## Acknowledgements

We thank Corinna H Engelhardt, Stefanie Kurth, and Jessica von Bongartz (RWTH-Aachen University) for excellent technical assistance. We are grateful to Pavel Talacko and Petr Zacek for Proteomic Core Facility support (BIOCEV, Charles University).

## Additional information

### Funding

| Funder | Grant reference number | Author |
|---|---|---|
| Deutsche Forschungsgemeinschaft | 368482240/GRK2416 | Marc Spehr |
| Deutsche Forschungsgemeinschaft | 378028035 | Yoram Ben-Shaul Marc Spehr |
| Volkswagen Foundation | I/83533 | Marc Spehr |
| German-Israeli Foundation for Scientific Research and Development | 1-1193-153.13/2012 | Yoram Ben-Shaul Marc Spehr |

The funders had no role in study design, data collection and interpretation, or the decision to submit the work for publication.

### Author contributions

Maximilian Nagel, Conceptualization, Resources, Data curation, Formal analysis, Investigation, Visualization, Methodology, Writing – original draft, Writing – review and editing; Marco Niestroj, Investigation; Rohini Bansal, Investigation, Writing – review and editing; David Fleck, Methodology, Writing – review and editing; Angelika Lampert, Visualization, Writing – review and editing; Romana Stopkova, Investigation, Methodology; Pavel Stopka, Conceptualization, Resources, Formal analysis, Supervision, Visualization, Methodology, Project administration, Writing – review and editing; Yoram Ben-Shaul, Conceptualization, Supervision, Funding acquisition, Project administration, Writing – review and editing; Marc Spehr, Conceptualization, Resources, Data curation, Supervision, Funding acquisition, Visualization, Methodology, Writing – original draft, Project administration, Writing – review and editing

### Author ORCIDs

David Fleck http://orcid.org/0000-0002-6692-2388
Angelika Lampert http://orcid.org/0000-0001-6319-6272
Pavel Stopka http://orcid.org/0000-0002-1104-1655
Yoram Ben-Shaul https://orcid.org/0000-0002-0407-4221
Marc Spehr https://orcid.org/0000-0001-6616-4196

### Ethics

Mice were maintained and sacrificed according to European Union legislation (Directive 2010/63/EU) and recommendations by the Federation of European Laboratory Animal Science Associations (FELASA). All experimental procedures were approved by the State Agency for Nature, Environment and Consumer Protection (LANUV).

Reviewer #1 (Public Review): https://doi.org/10.7554/eLife.90529.3.sa1

Reviewer #2 (Public Review): https://doi.org/10.7554/eLife.90529.3.sa2
Reviewer #3 (Public Review): https://doi.org/10.7554/eLife.90529.3.sa3
Author response https://doi.org/10.7554/eLife.90529.sa4

## Additional files

### Supplementary files
- MDAR checklist

### Data availability
All data is available in the main text or the supplementary materials. nLC-MS/MS proteomics data have been deposited to the ProteomeXchange Consortium via the PRIDE (Perez-Riverol et al., 2022) partner repository with the dataset identifier PXD042324. Metabolomics data have been deposited to the EMBL-EBI MetaboLights database with the identifier MTBLS7439.

The following datasets were generated:

| Author(s) | Year | Dataset title | Dataset URL | Database and Identifier |
|---|---|---|---|---|
| Stopka P | 2024 | Volatile urine metabolomes of the house mouse (*Mus musculus*) | https://www.ebi.ac.uk/metabolights/MTBLS7439 | MetaboLights, MTBLS7439 |
| Stopka P | 2024 | Urine proteomes of the house mouse | https://www.ebi.ac.uk/pride/archive/projects/PXD042324/ | PRIDE, PXD042324 |

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
