## [Editor Report · eLife assessment]

This carefully executed study provides a comparison of the chemical composition of mouse urine across strain and sex with the responses of vomeronasal sensory neurons, which are responsible for detecting chemical social cues. While the authors did not examine all molecular classes found in mouse urine or directly test whether the urinary volatile chemicals that vary with sex and strain are effective vomeronasal neuron ligands, **solid** data are provided that will be of significant interest to those studying chemical communication in rodents. This work should provide a **valuable** foundation for future research that will determine which molecules drive sex- and strain-specific vomeronasal responses.

---

## [Referee Report · Reviewer #1 (Public Review)]

In this manuscript, Nagel et al. sought to characterize the composition of urinary compounds, some of which are putative chemosignals. They used urines from adult males and females in three different strains, including one wild-derived strain. By performing mass spectrometry of two classes of compounds: volatile organic compounds and proteins, they found that urines from inbred strains are qualitatively similar to those of a wild strain. This finding is significant because there is a high degree of diversity in different inbred strains and wild mice, with respect to the polymorphisms of chemosensory receptor genes and expression of vomeronasal ligands previously identified. Notably, their study did not characterize steroids, which represent a major class of urinary chemosignals activating vomeronasal neurons. Therefore, important future studies should address the strain dependence of steroid composition in urines.

In the second part of this work, the authors used calcium imaging to monitor the pattern of vomeronasal neuron responses to these urines. By performing pairwise comparisons, the authors found a large degree of strain-specific response and a relatively minor response to sex-specific urinary stimuli. This is a finding generally in agreement with previous calcium imaging work by Ron Yu and colleagues in 2008. The authors extend the previous work by using urines from wild mice. They further report that the concentration diversity of urinary compounds in different urine batches is largely uncorrelated with the activity profiles of these urines. In addition, the authors found that the patterns of vomeronasal neuron response to urinary cues are not identical when measured using different recipient strains.

The pitfalls of this study are the omission of steroids for the mass spectrometry experiments and the indirect (correlational) nature of their mass spectrometry data and activity data. Whether the urinary compounds identified in this study activate vomeronasal neurons were not tested.

Nevertheless, the major contribution of this work is the identification of specific molecules in mouse urines. This work is likely to be of significant interest to researchers in chemosensory signaling in mammals and could provide a systematic avenue to exhaustively identify additional pheromones in mice.

---

## [Referee Report · Reviewer #2 (Public Review)]

This manuscript by Nagel et al provides a comprehensive examination of the chemical composition of mouse urine (an important source of semiochemicals) across strain and sex, and correlates these differences with functional responses of vomeronasal sensory neurons (an important sensory population for detecting chemical social cues). The strength of the work lies in the careful and comprehensive imaging and chemical analyses, the rigor of quantification of functional responses, and the insight into the relevance of olfactory work on lab-derived vs wild-derived mice.

With regards to the chemical analysis, the reader should keep in mind (and the authors acknowledge) that a difference in the concentration of a chemical across strain or sex does not necessarily mean that that chemical is used for chemical communication. In the most extreme case, the animals may be completely insensitive to the chemical. Thus, the fact that the repertoire of proteins and volatiles could potentially allow sex and/or strain discrimination, it is unclear to what degree both are used in different situations.

---

## [Referee Report · Reviewer #3 (Public Review)]

Summary:

The manuscript by Nagel, et al. describes studies of mouse vomeronasal sensory neuron (VSN) tuning to mouse urine samples across different sexes and strains, including wild mice, alongside mass spectrometry analysis of the same samples. The authors performed live Ca2+ imaging (CAL520 dye) of VSNs in acute vomeronasal organ (VNO) slices to determine how VSNs are tuned to pairs of stimuli that differ in their origin (e.g. male C57BL/6 versus male BALB/c urine, male C57BL/6 versus female C57BL/6, etc.). For each pair of tested odorants, the results measure the proportion of VSNs that respond to both stimuli ("generalists") or just one of the two ("specialists"), as well as metrics of tuning preference and response reliability. The authors find in most cases that generalists make up a larger proportion of responsive VSNs than specialists, but several pairwise comparisons showed a high degree of strain selectivity. Notably, the authors evaluated VSN tuning in both male C57BL/6 and male BALB/c VNOs, finding strain-dependent differences in the representation of mouse urine. Alongside these measurements of VSN tuning, the authors report results of mass spectrometry analyses of volatiles and proteins in the same urine samples. These analyses indicated a number of molecules in each category that vary across sex and strain, and therefore represent candidate vomeronasal ligands. However, this study did not directly test whether any of these candidate molecules drives VSN activity. Overall, this work provides solid information related to mouse vomeronasal chemosensation.

Strengths:

A strength of the current study is its focus on characterizing the neural responses of the VNO to urine derived from wild mice. The majority of existing vomeronasal system research has relied on the use of inbred strains for both neural response recordings and investigations of candidate vomeronasal system ligands. Inbreeding in laboratory environments may alter the chemical composition of bodily secretions, thereby potentially changing the information they contain. Moreover, the more homogeneous nature of inbred strains could be critical when studying the AOS mediated social aspects. If there exist noticeable differences in the chemical composition of secretions from wild animals compared to inbred strains, this would suggest that future research must consider natural sources of candidate ligands outside of inbred strains. This work identifies some intriguing differences, worthy of further exploration, between the urine composition of wild mice versus inbred mice, as well as disparities in how the VNO responds to urine from these different sources. However, the molecular composition and VNO responsiveness to wild mouse urine was found to be highly overlapping with inbred mouse urine, supporting the continued investigation of candidate ligands found in inbred mouse urine.

Another positive aspect of this work is its use of the same set of stimuli as a previous study by the same authors (Bansal et al., 2021) in the downstream accessory olfactory bulb. The consistency in stimulus selection facilitates a comparison of information processing of sex and strain information from the sensory periphery to the brain. Although comparisons between the two connected regions are not a focus of this work, and methodological differences (e.g., Ca2+ imaging versus electrophysiology) may introduce caveats into comparisons, the support of "apples to apples" comparisons across connected circuits is critical to progress in the field.

Finally, this study directly measured VSN tuning in both male C57BL/6 and male BALB/c VNOs, finding subtle but important differences in the representation of mouse urine in these two recipient strains. Given that there is a long history of behavioral research into strain-specific differences in social behavior, this research paves the way for future studies into how different mouse strains detect and process social chemosignals.

Weaknesses:

One of the primary objectives in this study is to ascertain the extent to which the response profiles of VSNs are specific to sex and strain. The design of these Ca2+ imaging experiments uses a simple stimulus design, using two interleaved bouts of stimulation with pairs of urine (e.g., male versus female C57BL/6, male C57BL/6 versus male BALB/c) at a single dilution factor (1:100). This introduces two significant limitations: (1) the "generalist" versus "specialist" descriptors pertain only to the specific pairwise comparisons made and (2) there is no information about the sensitivity/concentration-dependence of the responses.

The functional measurements of VSN tuning to various pairs of urine stimuli are presented alongside mass spectrometry-based comparisons. However, the mass spectrometry-based analysis was performed separately from VSN tuning experiments/analysis. The juxtaposition of these measurements may give some readers the impression that VSN tuning measurements were integrated with molecular profiling (i.e., that the molecular diversity was causally related physiological responses). This is a hypothesis raised by the parallel studies, but not a supported conclusion of the current work.

The impact of mass spectrometry findings is acknowledged to be limited to nonvolatile organic compounds and proteins/peptides, and that it is possible that few of these candidate molecules are active in the VNO. Moreover, it remains possible that the VSN responses are driven mostly by small nonvolatiles (e.g., polar steroids), a class of strong VSN ligands that were excluded from molecular analysis.

---

## [Author Response]

The following is the authors’ response to the original reviews.

General comments

All three experts have raised excellent ideas and made important suggestions to extend the scope of our study and provide additional information. While we fully acknowledge that these points are valid and would provide exciting new knowledge, we also should not lose track of the fact that a single study cannot cover all bases. Sulfated steroids, for example, are clearly essential components of mouse urine. Unfortunately, however, all chemical analysis approaches are limited and the one we opted for is not suitable for analysis of such signaling molecules. Future studies should certainly focus on these aspects. The same holds true for the fact that we do not know which of the identified compounds are actually VSN ligands. These are inherent limitations of the approach, and we are not claiming otherwise.

**Reviewer #1 (Public Review):**
(1) In this manuscript, Nagel et al. sought to comprehensively characterize the composition of urinary compounds, some of which are putative chemosignals. They used urines from adult males and females in three different strains, including one wild-derived strain. By performing mass spectrometry of two classes of compounds: volatile organic compounds and proteins, they found that urines from inbred strains are qualitatively similar to those of a wild strain. This finding is significant because there is a high degree of genetic diversity in wild mice, with chemosensory receptor genes harboring many polymorphisms.

We agree and thank the Reviewer for his / her positive assessment.

(2) In the second part of this work, the authors used calcium imaging to monitor the pattern of vomeronasal neuron responses to these urines. By performing pairwise comparisons, the authors found a large degree of strain-specific response and a relatively minor response to sex-specific urinary stimuli. This is a finding generally in agreement with previous calcium imaging work by Ron Yu and colleagues in 2008. The authors extend the previous work by using urines from wild mice. They further report that the concentration diversity of urinary compounds in different urine batches is largely uncorrelated with the activity profiles of these urines. In addition, the authors found that the patterns of vomeronasal neuron response to urinary cues are not identical when measured using different recipient strains. This fascinating finding, however, requires an additional control to exclude the possibility that this is not due to sampling error.

We thank Reviewer 1 for pointing this out. We agree that this is truly a “fascinating finding.” Reviewer 1 emphasizes that we need to add an “additional control to exclude […] that this is not due to sampling error”, and he / she elaborates on the required control in his / her Recommendations For The Authors (see below). Reviewer 1 states that “for Fig. 5, in order to conclude that the same urine activates a different population of VSNs in two different strains, a critical control is needed to demonstrate that this is not due to the sampling variability - as compositions of V1Rs and V2Rs could vary between different slices, one preferred control is to use VNO slices from the same strain and compare the selectivity used here across the A-P axis.” Importantly, we believe that this is already controlled for. In fact, for each experiment, we routinely prepare VNO slices along the organ’s entire anterior-to-posterior axis (not including the most anterior tip, where the VNO lumen tapers into the vomeronasal duct, and the most posterior part, the lumen ‘‘twists’’ toward the ventral aspect and its volume decreases (see Figs. 7 & S7 in Hamacher et al., 2024, Current Biology)). This usually yields ~7 slices per individual experiment / session. Therefore, we routinely sample and average across the entire VNO anterior-to-posterior axis for each experiment. In Fig. 5, in which we analyzed whether the “same urine activates a different population of VSNs in two different strains”, individual independent experiments from each strain (C57BL/6 versus BALB/c) amounted to (a) n = 6 versus n = 8; (b) n = 10 versus n = 10; (c) n = 7 versus n = 9; (d) n = 9 versus n = 10; (e) n = 10 versus n = 9; and (f) n = 12 versus n = 10. Together, we conclude that it is very unlikely that the considerably different response profiles measured in different recipient strains result from a “sampling error.”

To clarify this point in the revised manuscript, we now explain our sampling routine in more detail in the Materials and Methods. Moreover, we now also refer to this point in the Results.

(3) There are several weaknesses in this manuscript, including the lack of analysis of the compositions of sulfated steroids and other steroids, which have been proposed to be the major constituents of vomeronasal ligands in urines and the indirect (correlational) nature of their mass spectrometry data and activity data.

Reviewer 1 is correct to point out that our chemical profiling approach omits (sulfated) steroids. We are aware of this weakness. We deliberately decided to omit steroids as well as other nonvolatile small organic molecules for three main reasons: (i) as the reviewer points out, (sulfated) steroid composition has been the focus of analysis in several previous studies and there is ample published information available on their role as VSN stimuli; (ii) the analytical tools available to us do not allow comprehensive profiling of non-volatile small organic molecules; employing two-dimensional head-space GC-MS as well as LC-MS/MS is not suitable for steroid detection; and (iii) the relatively small sample volumes forced us to prioritize and focus on specific chemical classes (in our case, VOCs and proteins). We made an effort to use of the exact same stimuli as previously employed to investigate sensory representations in the accessory olfactory bulb (AOB) (Bansal et al., 2021), a feature that we consider a strength of the current study. However, this entailed that we had to effectively split our samples, further reducing the available sample volume.

We acknowledge that we did not sufficiently describe our rationale for focusing on VOCs and proteins on the previous version of the manuscript (nor did we discuss the known role of (sulfated) steroids in VSN signaling in adequate detail). We have now made an effort to address these shortcomings in the revised manuscript. Specifically, we have added new text to the Introduction (“Prominent molecularly identified VSN stimuli include various sulfated steroids (Celsi et al., 2012; Fu et al., 2015; Haga-Yamanaka et al., 2015, 2014; Isogai et al., 2011; Nodari et al., 2008; Turaga and Holy, 2012), which could reflect the dynamic endocrine state of an individual.”) and the Discussion (“Notably, our chemical profiling approach omits (sulfated) steroids other non-volatile small organic molecules, which have previously been identified in mouse urine as VSN stimuli (Nodari et al., 2008). Caution should thus be exerted to not attempt to fully explain VSN response specificity based on VOC and protein content alone.” & “In line with the notion of highly selective vomeronasal sampling is our observation that the concentration differences between compounds shared among strains, which are often substantial, are not reflected by similarly pronounced differences in response strength among generalist VSNs. There are several, not necessarily mutually exclusive explanations for this finding: First, concentration could simply not be a read-out parameter for VSNs, which would support previous ideas of concentration-invariant VSN activity (Leinders-Zufall et al., 2000). Second, the concentrations in freshly released urine could just exceed the dynamic tuning range of VSNs since, particularly for VOCs, natural signals (e.g., in scent marks) must be accessible to a recipient for a prolonged amount of time (sometimes days). A similar rationale could explain the increased protein concentrations in male urine, since male mice use scent marking to establish and maintain their territories and urinary lipocalins serve as long-lasting reservoirs of VOCs (Hurst et al., 1998). Third, generalist VSNs might sample information only from a select subset of urinary compounds, which, given their role as biologically relevant chemosignals, might be released at tightly controlled (and thus similar) concentrations. In fact, in the most extreme scenario, several compounds that do display substantial strain- and/or sex-specific differences in concentration might not act as chemosignals at all. Forth, to some extent, different response profiles could be attributed to non-volatile small organic molecules such as steroids (Nodari et al., 2008), which were beyond the focus of our chemical analysis.”).

(4) Overall, the major contribution of this work is the identification of specific molecules in mouse urines. This work is likely to be of significant interest to researchers in chemosensory signaling in mammals and provides a systematic avenue to exhaustively identify vomeronasal ligands in the future.

We thank the Reviewer for his / her generally positive assessment.

**Reviewer #2 (Public Review):**
(1) This manuscript by Nagel et al provides a comprehensive examination of the chemical composition of mouse urine (an important source of semiochemicals) across strain and sex, and correlates these differences with functional responses of vomeronasal sensory neurons (an important sensory population for detecting chemical social cues). The strength of the work lies in the careful and comprehensive imaging and chemical analyses, the rigor of quantification of functional responses, and the insight into the relevance of olfactory work on lab-derived vs wild-derived mice.

We thank the Reviewer for his / her generally positive assessment.

(2) With regards to the chemical analysis, the reader should keep in mind that a difference in the concentration of a chemical across strain or sex does not necessarily mean that that chemical is used for chemical communication. In the most extreme case, the animals may be completely insensitive to the chemical. Thus, the fact that the repertoire of proteins and volatiles could potentially allow sex and/or strain discrimination, it is unclear to what degree both are used in different situations.

Reviewer 2 is correct to point out that sex- and/or strain-dependent differences in urine molecular composition do not automatically attribute a signaling function to those molecules. We concur and, in fact, stress this point many times throughout the manuscript. In the Results, for example, we point out (i) that “in female urine, BALB/c-specific proteins are substantially underrepresented, a fact not reflected by VSN response profiles”, (ii) that “as observed in C57BL/6 neurons, the skewed distributions of protein concentration indices were not reflected by BALB/c generalist VSN profiles”, and (iii) that “VSN population response profiles do not reflect the global molecular content of urine, suggesting that the VNO functions as a rather selective molecular detector.” Moreover, in the Discussion, we state (i) that “caution should thus be exerted to not attempt to fully explain VSN response specificity based on VOC and protein content alone”; (ii) that, for several sex- and/or strain-specific molecules, none “has previously been attributed a chemosensory function. Challenging the mouse VNO with purified recombinant protein(s) will help elucidate whether such functions exist”; (iii) that “generalist VSNs might sample information only from a select subset of urinary compounds, which, given their role as biologically relevant chemosignals, might be released at tightly controlled (and thus similar) concentrations”; and (iv) that “to some extent, different response profiles could be attributed to non-volatile small organic molecules such as steroids (Nodari et al., 2008), which were beyond the focus of our chemical analysis.”

In the revised manuscript, we now aim to even more strongly emphasize the point made by Reviewer 2. In the Discussion, we have deleted a sentence that read: “Sex- and strain-specific chemical profiles give rise to unique VSN activity patterns.” Moreover, we have added the following statement: “In fact, in the most extreme scenario, several compounds that do display substantial strain- and/or sex-specific differences in concentration might not act as chemosignals at all.”

**Reviewer #3 (Public Review):**
(1) One of the primary objectives in this study is to ascertain the extent to which the response profiles of VSNs are specific to sex and strain. The design of these Ca2+ imaging experiments uses a simple stimulus design, using two interleaved bouts of stimulation with pairs of urine (e.g. male versus female C57BL/6, male C57BL/6 versus male BALB/c) at a single dilution factor (1:100). This introduces two significant limitations: (1) the "generalist" versus "specialist" descriptors pertain only to the specific pairwise comparisons made and (2) there is no information about the sensitivity/concentration-dependence of the responses.

Reviewer 3 points to two limitations of our VSN activity assay. He / she is correct to mention that characterizing a VSN as generalist or specialist based on a “pairwise comparison” should not be the basis of attributing such a “generalist” or “specialist” label in general (i.e., regarding the global stimulus space). We acknowledge this point, but we do not regard this as a limitation of our study since we are not investigating rather broad (i.e., multidimensional) questions of selectivity. All we are asking in the context of this study is whether VSNs - when being challenged with pairs of sex- or strain-specific urine samples - act as rather selective semiochemical detectors. Of course, one can always think of a study design that provides more information. However, we here opted for an assay that - in our hands - is robust, “low noise” (i.e., displays low intrinsic signal variability as evident form reliability index calculations), ensures recovery from VSN adaptation (Wong et al., 2018), and, importantly, answers the specific question we are asking.

Regarding the second point (“there is no information about the sensitivity/concentrationdependence of the responses”), we would like to emphasize that this was not a focus of our study either. In fact, concentration-dependence of VSN activity has been a major focus of several previous studies referenced in our manuscript (e.g., Leinders-Zufall et al., 2000; He et al., 2008), albeit with contradictory results. In our study, we ask whether a pair of stimuli that we have shown to display, in part, strikingly different chemical composition (both absolute and relative) preferentially activates the same or different VSNs. With this question in mind, we believe that our assay (and its results) are highly informative.

(2) The functional measurements of VSN tuning to various pairs of urine stimuli are consistently presented alongside mass spectrometry-based comparisons. Although it is clear from the manuscript text that the mass spectrometry-based analysis was separated from the VSN tuning experiments/analysis, the juxtaposition of VSN tuning measurements with independent molecular diversity measurements gives the appearance to readers that these experiments were integrated (i.e., that the diversity of ligands was underlying the diversity of physiological responses). This is a hypothesis raised by the parallel studies, not a supported conclusion of the work. This data presentation style risks confusing readers.

As Reviewer 3 points out correctly “it is clear from the manuscript text that the mass spectrometry-based analysis was separated from the VSN tuning experiments/analysis.” In the figures, we try make the distinction between VSN response statistics and chemical profiling more obvious by gray shadows that link the plots depicting VSN response characteristics to the general pie charts.

We now also made an extra effort to avoid “confusing readers” by stating in the Discussion (i) that “caution should thus be exerted to not attempt to fully explain VSN response specificity based on VOC and protein content alone”; (ii) that, for several sex- and/or strain-specific molecules, none “has previously been attributed a chemosensory function. Challenging the mouse VNO with purified recombinant protein(s) will help elucidate whether such functions exist”; (iii) that “generalist VSNs might sample information only from a select subset of urinary compounds, which, given their role as biologically relevant chemosignals, might be released at tightly controlled (and thus similar) concentrations”; and (iv) that “to some extent, different response profiles could be attributed to non-volatile small organic molecules such as steroids (Nodari et al., 2008), which were beyond the focus of our chemical analysis.” Moreover, we have deleted a sentence that read: “sex- and strain-specific chemical profiles give rise to unique VSN activity patterns”, and we have added the following statement: “In fact, in the most extreme scenario, several compounds that do display substantial strain- and/or sex-specific differences in concentration might not act as chemosignals at all.”

However, we believe that there is value in presenting “VSN tuning measurements” next to “independent molecular diversity measurements.” While these are independent measurements, their similarity or, quite frequently, lack thereof are informative. We are sure that by taking the above “precautions” we have now mitigated the risk of “confusing readers.”

(3) The impact of mass spectrometry findings is limited by the fact that none of these molecules (in bulk, fractions, or monomolecular candidate ligands) were tested on VSNs. It is possible that only a very small number of these ligands activate the VNO. The list of variably expressed proteins - especially several proteins that are preferentially found in female urine - is compelling, but, again, there is no evidence presented that indicates whether or not these candidate ligands drive VSN activity. It is noteworthy that the largest class of known natural ligands for VSNs are small nonvolatiles that are found at high levels in mouse urine. These molecules were almost certainly involved in driving VSN activity in the physiology assays (both "generalist" and "specialist"), but they are absent from the molecular analysis.

Reviewer 3 is right, of course, that at this point we have not tested the identified molecules on VSNs. This is clearly beyond the scope of the present study. We believe that the data we present will be the basis of (several full-length) future studies that aim to identify specific ligands and - best case scenario - receptor-ligand pairs. We find it hard to concur that our study, which provides the necessary basis for those future endeavors, is regarded as “incomplete”. By design, all studies are somewhat incomplete, i.e., there are always remaining questions and we are not contesting that.

It is true, of course, that a class of “known natural ligands for VSNs are small nonvolatiles.” As we replied above, our chemical profiling approach omits (sulfated) steroids. We are aware of this weakness. We deliberately decided to omit steroids as well as other non-volatile small organic molecules for three main reasons: (i) steroid composition has been the focus of analysis in several previous studies and there is ample published information available on their role as VSN stimuli; (ii) the analytical tools available to us do not allow comprehensive profiling of non-volatile small organic molecules; employing two-dimensional head-space GC-MS as well as LC-MS/MS is not suitable for steroid detection; and (iii) the relatively small sample volumes forced us to prioritize and focus on specific chemical classes (in our case, VOCs and proteins). We made an effort to use of the exact same stimuli as previously employed to investigate sensory representations in the accessory olfactory bulb (AOB) (Bansal et al., 2021), a fact that we consider a key strength of our current study. However, this entailed that we had to effectively split our samples, further reducing the available sample volume.

We acknowledge that we did not sufficiently describe our rationale for focusing on VOCs and proteins on the previous version of the manuscript (nor did we discuss the known role of (sulfated) steroids in VSN signaling in adequate detail). We have now made an effort to address these shortcomings in the revised manuscript. Specifically, we have added new text to the Introduction (“Prominent molecularly identified VSN stimuli include various sulfated steroids (Celsi et al., 2012; Fu et al., 2015; Haga-Yamanaka et al., 2015, 2014; Isogai et al., 2011; Nodari et al., 2008; Turaga and Holy, 2012), which could reflect the dynamic endocrine state of an individual.”) and the Discussion (“Notably, our chemical profiling approach omits (sulfated) steroids other non-volatile small organic molecules, which have previously been identified in mouse urine as VSN stimuli (Nodari et al., 2008). Caution should thus be exerted to not attempt to fully explain VSN response specificity based on VOC and protein content alone.” & “In line with the notion of highly selective vomeronasal sampling is our observation that the concentration differences between compounds shared among strains, which are often substantial, are not reflected by similarly pronounced differences in response strength among generalist VSNs. There are several, not necessarily mutually exclusive explanations for this finding: First, concentration could simply not be a read-out parameter for VSNs, which would support previous ideas of concentration-invariant VSN activity (Leinders-Zufall et al., 2000). Second, the concentrations in freshly released urine could just exceed the dynamic tuning range of VSNs since, particularly for VOCs, natural signals (e.g., in scent marks) must be accessible to a recipient for a prolonged amount of time (sometimes days). A similar rationale could explain the increased protein concentrations in male urine, since male mice use scent marking to establish and maintain their territories and urinary lipocalins serve as long-lasting reservoirs of VOCs (Hurst et al., 1998). Third, generalist VSNs might sample information only from a select subset of urinary compounds, which, given their role as biologically relevant chemosignals, might be released at tightly controlled (and thus similar) concentrations. In fact, in the most extreme scenario, several compounds that do display substantial strain- and/or sex-specific differences in concentration might not act as chemosignals at all. Forth, to some extent, different response profiles could be attributed to non-volatile small organic molecules such as steroids (Nodari et al., 2008), which were beyond the focus of our chemical analysis.”).

**Reviewer #1 (Recommendations For The Authors):**
(1) I find that the study is highly valuable for researchers in this field. With the finding that wild mouse urines do not elicit significantly more variable responses from urines from inbred strains, researchers can now be reassured to use inbred strains to gain general insights on pheromone signaling.A major omission of this study is non-volatile small organic molecules such as steroids. These compounds are the only molecular class in urine that have been identified to stimulate specific vomeronasal receptors to date. It is unclear to me that the specificity of VOC and proteins can alone fully explain the response specificity of the VSNs that have been monitored in this study. The discussion of this topic is highly beneficial for the readers.

Reviewer 1 is correct to point out that our chemical profiling approach omits (sulfated) steroids. We are aware of this weakness. We deliberately decided to omit steroids as well as other nonvolatile small organic molecules for three main reasons: (i) as the reviewer points out, (sulfated) steroid composition has been the focus of analysis in several previous studies and there is ample published information available on their role as VSN stimuli; (ii) the analytical tools available to us do not allow comprehensive profiling of non-volatile small organic molecules; employing two-dimensional head-space GC-MS as well as LC-MS/MS is not suitable for steroid detection; and (iii) the relatively small sample volumes forced us to prioritize and focus on specific chemical classes (in our case, VOCs and proteins). We made an effort to use of the exact same stimuli as previously employed to investigate sensory representations in the accessory olfactory bulb (AOB) (Bansal et al., 2021), a fact that we consider a key strength of our current study. However, this entailed that we had to effectively split our samples, further reducing the available sample volume.

We acknowledge that we did not sufficiently describe our rationale for focusing on VOCs and proteins on the previous version of the manuscript (nor did we discuss the known role of (sulfated) steroids in VSN signaling in adequate detail). We have now made an effort to address these shortcomings in the revised manuscript. Specifically, we have added new text to the Introduction (“Prominent molecularly identified VSN stimuli include various sulfated steroids (Celsi et al., 2012; Fu et al., 2015; Haga-Yamanaka et al., 2015, 2014; Isogai et al., 2011; Nodari et al., 2008; Turaga and Holy, 2012), which could reflect the dynamic endocrine state of an individual.”) and the Discussion (“Notably, our chemical profiling approach omits (sulfated) steroids other non-volatile small organic molecules, which have previously been identified in mouse urine as VSN stimuli (Nodari et al., 2008). Caution should thus be exerted to not attempt to fully explain VSN response specificity based on VOC and protein content alone.” & “In line with the notion of highly selective vomeronasal sampling is our observation that the concentration differences between compounds shared among strains, which are often substantial, are not reflected by similarly pronounced differences in response strength among generalist VSNs. There are several, not necessarily mutually exclusive explanations for this finding: First, concentration could simply not be a read-out parameter for VSNs, which would support previous ideas of concentration-invariant VSN activity (Leinders-Zufall et al., 2000). Second, the concentrations in freshly released urine could just exceed the dynamic tuning range of VSNs since, particularly for VOCs, natural signals (e.g., in scent marks) must be accessible to a recipient for a prolonged amount of time (sometimes days). A similar rationale could explain the increased protein concentrations in male urine, since male mice use scent marking to establish and maintain their territories and urinary lipocalins serve as long-lasting reservoirs of VOCs (Hurst et al., 1998). Third, generalist VSNs might sample information only from a select subset of urinary compounds, which, given their role as biologically relevant chemosignals, might be released at tightly controlled (and thus similar) concentrations. Forth, to some extent, different response profiles could be attributed to non-volatile small organic molecules such as steroids (Nodari et al., 2008), which were beyond the focus of our chemical analysis.”).

(2) How many different wild mouse urines were tested in this study? Is this sufficient to capture the diversity of wild *M. musculus* in local (Prague) habitats?

We thank the reviewer for pointing this out. For the present study, 20 male (M) and 27 female (F) wild mice were caught at six different sites in the broader Prague area (i.e., Bohnice 50.13415N, 14.41421E; 2M+4F), Dolni Brezany (49.96321N, 14.4585E; 3M+4F), Hodkovice (49.97227N, 14.48039E; 5M+6F), Písnice (49.98988N, 14.46625E; 3M+6F), Lhota (49.95369N, 14.43087E; 1M+2F), and Zalepy (49.9532N, 14.40829E; 6M+5F). 18 of the 27 wild females were caught pregnant. The remaining 9 females were mated with males caught at the same site and produced offspring within a month. When selecting 10 male and 10 female individuals from first-generation offspring for urine collection, we ensured that all six capture sites were represented and that age-matched animals displayed similar weight (~17g). We believe that this capture / breeding strategy sufficiently represents “the diversity of wild *M. musculus* in local (Prague) habitats.” In the revised manuscript, we have now included these details in the Materials and Methods.

(3) I found Figure 1e and figures in a similar format confusing - one panel describes the response statistics of VSNs, and other panels show the number of compounds found in different MS profiling, which is not immediately obvious from the figures. Is the y-axis legend correct (%)?

We now try make the distinction between VSN “response statistics” and chemical profiling more obvious by gray shadows that link the plots depicting VSN response characteristics to the general pie charts. Moreover, we thank the Reviewer for pointing out the mislabeling of the y-axis. Accordingly, we have deleted “%” in all corresponding figures.

(4) For Figure 5, in order to conclude that the same urine activates a different population of VSNs in two different strains, a critical control is needed to demonstrate that this is not due to the sampling variability - as compositions of V1Rs and V2Rs could vary between different slices, one preferred control is to use VNO slices from the same strain and compare the selectivity used here across the A-P axis.

We thank Reviewer 1 for pointing this out. Importantly, we believe that this is already controlled for (see our response to the Public Review). In fact, for each experiment, we routinely prepare VNO slices along the entire anterior-to-posterior axis (not including the most anterior tip, where the VNO lumen tapers into the vomeronasal duct, and the most posterior part, the lumen ‘‘twists’’ toward the ventral aspect and its volume decreases see Figs. 7 & S7 in Hamacher et al., 2024, Current Biology). This usually yields ~7 slices per individual experiment / session. Therefore, we routinely sample and average across the entire VNO anterior-to-posterior axis for each experiment. In Fig. 5, individual independent experiments from each strain (C57BL/6 versus BALB/c) amounted to (a) n = 6 versus n = 8; (b) n = 10 versus n = 10; (c) n = 7 versus n = 9; (d) n = 9 versus n = 10; (e) n = 10 versus n = 9; and (f) n = 12 versus n = 10. Together, we can thus exclude that the considerably different response profiles that we measured using different recipient strains result from a “sampling error.”

To clarify this point in the revised manuscript, we now explain our sampling routine in more detail in the Materials and Methods. Moreover, we now also mention this point in the Results.

**Reviewer #2 (Recommendations For The Authors):**
(1) Pg 5 Lines 3-16: This summary paragraph contains too much detail given that the reader has not read the paper yet, which makes it bewildering. This should be condensed.

We agree and have substantially condensed this paragraph.

(2) Pg 6 Line 5-8: This summary of the experimental design is obtuse and should be edited for clarity.

We have edited the relevant passage for clarity.

(3) Pg 6 Line 11: "VSNs were categorized..." Specialist vs generalist is defined as responding to one or both stimuli. This definition is placed right after saying that the cells were also tested with KCl. The reader might think that specialist vs generalist was defined in relation to KCl.

We have edited this sentence, which now reads: “Dependent on their individual urine response profiles, VSNs were categorized as either specialists (selective response to one stimulus) or generalists (responsive to both stimuli).”

(4) Pg 6 Line 13: "we recorded urine-dependent Ca2+ signals from a total of 16,715 VSNs". Is a "signal" a response? Did all 16,715 VSNs respond to urine? What was the total of KCl responsive cells recorded?

We edited the corresponding passage for clarification. The text now reads: “Overall, we recorded >43,000 K+-sensitive neurons, of which a total of 16,715 VSNs (38.4%) responded to urine stimulation. Of these urine-sensitive neurons, 61.4% displayed generalist profiles, whereas 38.6% were categorized as specialists (Figure 1c,d).”

(5) Pg 7 Line 6: The repeated use of the word "pooled" is confusing as it suggests a variation in the experiment. The authors should establish once in the Methods and maybe in the Results that stimuli were pooled across animals. Then they should just refer to the stimulus as male or female or BALB/c rather than "pooled" male etc.

We acknowledge the reviewer’s argument. Accordingly, we now introduce the experimental use of pooled urine once in the Methods and in the introductory paragraph of the Results. All other references to “pooled” urine in the Results and Captions have been deleted.

(6) Pg 7 Line 10: "...detected in >=3 out of 10 male..." For the chemical analysis, were these samples not pooled?

Correct. We deliberately did not pool samples for chemical analysis, but instead analyzed all individual samples separately (i.e., 60 samples were subjected to both proteomic and metabolomic analyses). Thus, the criterion that a VOC or protein must be detected in at least 3 of the 10 individual samples from a given sex/strain combination for a ‘present’ call (and in at least 6 of the 10 samples to be called ‘enriched’) ensures that the molecular signatures we identify are not “contaminated” by unusual aberrations within single samples.

For clarification, we now explicitly outline this procedure in the Methods (Experimental Design and Statistical Analysis – Proteomics and metabolomics).

(7) Pg 7 Line 23: In line 7, the specialist rate was defined as 5% in reference to the total KCl responsive cells. Here the specialist rate is defined from responsive cells. This is confusing.

We apologize for the confusion. In both cases, the numbers (%) refer to all K+-sensitive neurons. We have added this information to both relevant sentences (l. 7 as well as ll. 23-24). Note that the rate in ll. 23-24 refers to generalists.

(8) Pg 7 Line 25: Concentration index should be defined before its use here.

We have revised the corresponding sentence, which now reads: “By contrast, analogously calculated concentration indices (see Materials and Methods) that can reflect potential disparities are distributed more broadly and non-normally (Figure 1h).”

(9) Pg 7 Line 29: change "trivially" to "simply".

Done

(10) Pg 7 Line 30: What is meant by a "generalist" ligand? The neurons are generalists.Probably should read "common ligands"

We have changed the text accordingly.

(11) Pg 7 Line 31: What is meant by "global observed concentration disparities" ?

We have changed the text to “…represented by the observed general concentration disparities.”

(12) Pg 8 Lines 7-11: This section needs to be edited for clarity as it is very difficult to follow. For example, the definition of "enriched" is buried in a parenthetical. Also, it is very difficult to figure out what a "sample" is in this paper. Is it a pooled stimulus, or is it urine from an individual animal?

We apologize for the confusion. Throughout the paper a “sample” is a pooled stimulus (from all 10 individuals of a given sex/strain combination) for all physiological experiments. For chemical analysis a “sample” refers to urine from an individual animal.

(13)Pg 8 Line 11: "abundant proteins" Does this mean absolute concentration or enriched in one sample vs another?

We changed the term “abundant” to “enriched” as this descriptor has been defined (present in ≥6 of 10 individual samples) in the previous sentence.

(14) Pg 8 Line 18: "While 32.9% of all..." Please edit for clarity. What is the point?

The main point here is that, for VOCs, the vast majority of compounds (91.3%) are either generic mouse urinary molecules or are sex/strain-specific.

(15) Pg 10 Line 18: "Increased VSN selectivity..." This title is misleading as it suggests a change in sensitivity with animal exposure. I think the authors are trying to say "VSNs are more selective for strain than for sex". The authors should avoid the term "exposure to" when they mean "stimulation with" as the former suggests chronic exposure prior to testing.

We thank the reviewer for the advice and have changed the title accordingly. We also edited the text to avoid the term "exposure to" throughout the manuscript.

(16) Pg 12 Line 10: "we recorded hardly any..." Hardly any in comparison to what? BALB/c?

We apologize for the confusion. We have edited the text for clarity, which now reads: “In fact, (i) compared to an average specialist rate of 11.2% ± 6.6% (mean ± SD) calculated over all 13 binary stimulus pairs (n = 26 specialist types), we observed only few specialist responses upon stimulation with urine from wild females (2% and 3%, respectively), and…”

**Reviewer #3 (Recommendations For The Authors):**
(1) Related to the pairwise stimulus-response experimental design and analysis: there is precedent in the field for studies that explore the same topic (sex- and strain-selectivity), but measure VSN sensitivity across many urine stimuli, not just two at a time. This has been done both in the VNO (He et al, Science, 2008; Fu, et al, Cell, 2015) and in the AOB (Tolokh, et al, Journal of Neuroscience, 2013). The current manuscript does not cite these studies.

Reviewer 3 is correct and we apologize for this oversight. We now cite the two VSN-related studies by He et al. and Fu et al. in the Introduction.

(2) The findings of the mass spectrometry-based profiling of mouse urine - especially for volatiles - is only accessible through repositories, making it difficult to for readers to understand which molecules were found to be highly divergent between sexes/strains. There is value in the list of ligands to further investigate, but this information should be made more accessible to readers without having to comb through the repositories.

We agree that there “is value in the list of ligands to further investigate” and, accordingly, we now provide a table (Table 1) that lists the top-5 VOCs that – according to sPLS-DA – display the most discriminative power to classify samples by sex (related to Figure 2c) or strain (related to Figure 2d). For ease of identification, all entries list internal mass spectrometry identifiers, identifiers extracted from MS analysis database, the sex or strain that drives separation, which two-dimensional component / x-variate represents the most discriminative variable, PubChem chemical formula, PubChem common or alternative names, Chemical Entities of Biological Interest or PubChem Compound Identification, and the VOC’s putative origin.

(3) There is a long precedent for integrating molecular assessments and physiological recordings to identify specific ligands for the vomeronasal system:nonvolatiles (e.g., Leinders-Zufall, et al., Nature, 2000)peptides (e.g., Kimoto et al., Nature, 2005; Leinders-Zufall et al. Science, 2004; Riviere et al., Nature, 2009; Liberles, et al., PNAS, 2009)proteins (e.g., Chamero et al., Nature, 2007; Roberts et al., BMC Biology, 2010)excreted steroids and bile acids (Nodari et al., Journal of Neuroscience, 2008; Fu et al., Cell, 2015; Doyle, et al., Nature Communications, 2016)The Leinders-Zufall (2000), Roberts, and Nodari papers are referenced, but the broader efforts by the community to find specific drivers of vomeronasal activity are not fully represented in the manuscript. The focus of this paper is fully related to this broader effort, and it would be appropriate for this work to be placed in this context in the introduction and discussion.

We now refer to all of the studies mentioned in the Introduction (except the article published by Liberles et al. in 2009, since the authors of that study do not identify vomeronasal ligands).

(4) Throughout the manuscript (starting in Fig. 1h) the figure panels and captions use the term "response index" whereas the methods define a "preference index." It seems to be the case that these two terms are synonymous. If so, a single term should be consistently used. If not, this needs to be clarified.

We now consistently use the term “response index” throughout the manuscript.

(5) It would be useful to provide a table associated with Figure 2 - figure supplement 1 that lists the common names and/or chemical formulas for the volatiles that were found to be of high importance.

We agree and, accordingly, we now provide a table (Table 2) that lists VOC, which – according to Random Forest classification and resulting Gini importance scores – display the most discriminative power to classify samples by sex (related to Figure 2 - figure supplement 1a) or strain (related to Figure 2 - figure supplement 1b). Notably, it is generally reassuring that several VOCs are listed in both Table 1 and Table 2, emphasizing that two different supervised machine learning algorithms (i.e., sPLS-DA (Table 1) and Random Forest (Table 2)) yield largely congruent results.

(5) The use of the term "comprehensive" for the molecular analysis is a little bit misleading, as volatiles and proteins are just two of the many categories of molecules present in mouse urine.

We have now deleted most mentions of the term "comprehensive" when referring to the molecular analysis.

(7) Page 11, lines 24-27: The sentences starting "We conclude..." and ending in "semiochemical concentrations." These two sentences do not make sense. It is not known how many of the identified proteins are actual VSN ligands. Moreover, there is abundant evidence from other studies that individual VSN activity provides information about distinct semiochemical concentrations.

We have substantially edited and rephrased this paragraph to better reflect that different scenarios / interpretations are possible. The relevant text now reads: “We conclude that VSN population response strength might not be so strongly affected by strain-dependent concentration differences among common urinary proteins. In that case, it would appear somewhat unlikely that individual VSN activity provides fine-tuned information about distinct semiochemical concentrations. Alternatively, as some (or even many) of the identified proteins could not serve as vomeronasal ligands at all, generalist VSNs might sample information from only a subset of compounds which, in fact, are secreted at roughly similar concentrations.”

(8) The explanation of stimulus timing is mentioned several times but not defined clearly in methods. Page 19, lines 14-19 have information about the stimulus delivery device, but it would be helpful to have stimulus timing explicitly stated.

In addition to the relevant captions, we now explicitly state stimulus timing (i.e., 10 s stimulations at 180 s inter-stimulus intervals) in the Results.

(9) Typos:Page 10, line 7: "male biased" → "male-biased" for clarityWilcoxon "signed-rank" test is often misspelled "Wilcoxon singed ranked test" or "Wilcoxon signed ranked test"In the Fig. 3 legend, the asterisk meaning is unspecified."(im)balances" → imbalances (page 27, line 24; page 37, line 16; page 38, line 16)Figure 2 - figure supplement 1 and in Figure 2 - figure supplement 2, in the box-andwhisker plots the units are not specified in the graph or legend.”

We have made all required corrections.